# Quantum Shannon Information Theory—Design of Communication, Ciphers, and Sensors

**DOI:** 10.3390/e27111158

**Published:** 2025-11-14

**Authors:** Osamu Hirota

**Affiliations:** 1Quantum ICT Research Institute, Tamagawa University, 6-1-1, Tamagawa-Gakuen, Machida 194-8610, Japan; hirota@lab.tamagawa.ac.jp; 2Research and Development Initiative, Chuo University, 1-13-27, Kasuga, Bunkyou-ku, Tokyo 112-8551, Japan

**Keywords:** quantum technology, secure global optical network, reaction control system

## Abstract

One of the key aspects of Shannon theory is that it provides guidance for designing the most efficient systems, such as minimizing errors and clarifying the limits of coding. This theory has seen great developments in the 50 years since 1948. It has played a vital role in enabling the development of modern ultra-fast, stable, and highly dependable information and communication systems. Shannon theory is supported by statistical communication theories such as detection and estimation theory. The theory of communication systems that transmit Shannon information using quantum media is called quantum Shannon information theory, and research began in the 1960s. The theoretical formulation comparable to conventional Shannon theory has been completed. Its important role is to suggest that application of quantum effects will surpass existing communication performance. It would be meaningless if performance, efficiency, and utility were to deteriorate due to quantum effects, even if a certain new function is given. This paper suggests that there are various limitations to utilizing quantum Shannon information theory to benefit real-world communication systems and presents a theoretical framework for achieving the ultimate goal. Finally, we present the perfect secure cipher that overcomes the Shannon impossibility theorem without degrading communication performance and sensors as an example.

## 1. Introduction

In the 1970s, while upholding the MIT tradition, R.S. Kennedy and H.A. Haus of MIT shifted communications science toward a research direction that incorporates fundamental theories of communications that take quantum effects into account (Appendix A). Kennedy took the lead in this direction and began nurturing young researchers, which has resulted in research that has had a global impact in a variety of fields. The current boom in quantum information science can be traced back to their leadership.

In this field, theory for systems that transmit conventional information using quantum states is called quantum Shannon information theory. In order to construct such a theory, probability theory is needed to describe quantum mechanical phenomena in an information-theoretic way. It is formulated as quantum probability theory, which has a different mathematical system from conventional probability theory and can faithfully describe quantum mechanical phenomena. For these reasons, it is necessary to build communication theory for the above communication systems based on quantum probability theory. It is called quantum communication theory (or quantum detection and estimation theory) and was developed by Helstrom [1], Holevo [2], Yuen [3], and other researchers.

Like conventional Shannon theory, the construction of quantum Shannon information theory started based on quantum communication theory, the studying of the properties of Shannon mutual information in systems with quantum effects. The most important achievement of the evolution from quantum communication theory to quantum Shannon information theory is the proof of Shannon’s second theorem (channel coding theorem) for quantum communication systems. First, Hausladen et al. of the Jozsa group proved it by skillfully using the properties of random coding and quantum measurement on a typical subspace of the Hilbert space spanned by pure signal quantum states for the case where the quantum state communication channel is noiseless [4].

Following from this result, Holevo [5] and Schumacher-Westmoreland [6] proved that the discrete channel capacity of a general quantum channel with noise is given by Holevo information. This is now called the Holevo–Schumacher-Westmoreland theorem.

It is well known that the theorems on the coding limit are the most important ones in the application of information theory among the major theorems in information theory. In particular, the channel coding theorem shows that there exists a coding scheme that is error-free for an infinitely long typical sequence and that the information rate is ultimately equal to the channel capacity.

In response to such a limiting existence theorem, Gallager completed a theory that does not use a typical sequence and gives a limit to the average error probability of code words for a finite-length code system. This allows us to know the limiting characteristics of errors for a finite-length code system for any information rate up to the channel capacity when a channel matrix is given. This is called the theory of reliability functions, and it is well known that it made a major contribution to the process of constructing Shannon theory [7].

Thus, as a natural evolution of research, Holevo’s group introduced a reliability function corresponding to quantum communication channels following Gallager’s concept [8,9]. In addition, Helstrom, C. Bendjaballa et al. [10,11,12], and M. Ban et al. [13,14] formulated the cut-off rate of quantum systems. These trials are applications of Gallager theory to quantum systems. Just as Gallager’s reliability function theory was effective in analyzing communication systems following classical physics, the quantum reliability function and cut-off rate are also considered to be effective in analyzing communication systems with quantum effects. Based on this perspective, Holevo has completed a unified mathematical foundation for quantum systems [15].

In this paper, we summarize the development from quantum communication theory to quantum Shannon information theory in Section 2, Section 3, Section 4 and Section 5 and explain its applications to real-world communication technologies, showing quantum advantage for communication, ciphers, and sensors. The most dramatic result of this theory is the discovery of quantum stream ciphers, which solve the Shannon impossibility theorem for symmetric key ciphers and realize perfectly secure ciphers with short keys.

## 2. General Quantum Communication Channel Model

In this section, we denote the theoretical structure of general quantum systems and provide an explanation of the mathematical theory of quantum communication channels. In quantum theory for transmission of the information defined by Shannon, Shannon information as a finite set of alphabets is mapped to a quantum state. Therefore, the overall communication channel consists of a quantum state-to-quantum state communication channel and a quantum measurement channel that converts the received quantum state into a classical signal via measurement (Figure 1).

In order to analyze them as information theory, it is necessary to set up a communication channel model. A communication channel that transmits a quantum state onto which the signal is mapped is called a quantum state transmission channel. This channel is described by a completely positive map in the general mathematical sense [15].

**Definition** **1.**
*Let A,B be two *-algebras and S⊂A be an operator system. A linear map Φ:S→B is called completely positive if $\sum_{i,j=1}^{n}b_{i}^{*}\Phi \left(a_{i}^{*}a_{j}\right)b_{j}\in B$ is positive for all n∈N and for all $a_{i}\in S,b_{i}\in B$. The space of all such maps is denoted by CP(S,B).*


The key issue is the physical reality of such abstract descriptions. In Shannon theory, a channel is considered noiseless when the relationship between the input and output symbols is injective. From this perspective, it is important that the quantum state carrying Shannon information does not lose its quantum properties along the channel. Fortunately, the following theorem has been shown for real-world channels by Helstrom [16].

**Theorem** **1**({Helstrom})**.**
*In the transmission of quantum states through an optical communication channel with energy loss, the quantum state that can maintain a pure state is a coherent state which is defined by a|α>=α|α>, where a is a photon annihilation operator.*

This theorem has an affinity with modern communication technology. That is, the maximum capacity of reliable information transmission in general optical channels such as fiber and free space is achieved by a coherent state, considering the channel characteristics, the bandwidth of the light source, and the receiver.

In fact, in optical fiber communication, energy loss is the main issue. When the transmitting quantum state is a coherent state, which is the quantum state of general laser light, it maintains a pure state even if there is any loss, so it is treated as noiseless in the sense of quantum state transmission.

In free-space optical communication, in addition to energy loss, external noise is added, so the quantum state transmission channel becomes a noisier channel (Figure 2). But still, the coherent state provides the maximum capacity, which is discussed in Section 5.

**Remark** **1.**
*When a single photon (number state), a squeezed state, quantum entangled states, etc., are used as the transmitter state, energy loss or decoherence transforms these states into a mixed state, and the channel becomes non-injective. Thus, light sources of quantum states with strong quantum properties cannot meet the performance requirements of modern communication, which requires over 100 Gbit/s.*


Next, it is necessary to describe the process of identifying the quantum state that has arrived at the receiver. In this quantum measurement process, unavoidable noise due to the most basic observation operation in quantum mechanics appears. If this is modeled as quantum noise, it can be modeled as a quantum measurement communication channel. In other words, unavoidable errors occur in the identification of quantum states, and an optimization problem arises regarding their decision.

This type of model was proposed by Helstrom in 1967 and is called quantum detection and estimation theory [1]. The noise model of the photoelectric conversion process in the receiver of laser optical communication is the simplest physical example of quantum noise. In order to describe this quantum measurement channel, quantum probability theory, which describes the quantum measurement process, is required. Below we briefly present the mathematics of quantum probability as a basis for the following discussion.

## 3. Mathematical Foundation of Generalized Quantum Measurement and Decision Operator

### 3.1. Mathematical Formulae of Generalized Quantum Measurement

The mathematical structure of quantum mechanics is given based on the Hilbert space theory constructed by von Neumann. The representation of the quantum measurement process is formulated using the eigen state and eigen value of a self-adjoint operator. This is defined as follows.

**Definition** **2.**
*Let us assume that the quantum system has the self-adjoint operator as physical observable T and its quantum state ρ. The Born rule in the standard quantum measurement of the observable is described as follows:*

(1)
T|x>=x|x>,P(x)=Trρ|x><x|

*where |x><x| is called projection valued measure.*


The set of density operator ρ is a convex set, its extreme points being the one-dimensional projection. The corresponding states are called pure. Any measurement with values in the real number space $R_{M}$ is described by an affine map of the set of the states (density operator) to the set of probability distributions on $R_{M}$.

However, P. Benioff and others have been discussing the mathematical generalization of the projection process to describe the diversity of measurement processes. Finally the generalized quantum measurement process is described in the following form [17].

**Definition** **3.**
*Let us consider a generalized resolution of identity $\left\{X\left(r\right);r\in R_{M}\right\}$, i.e., the collections of Hermitian operators, satisfying $\sum_{r\in R_{M}}X\left(r\right)=I,X\left(r\right)\geq 0$. TrρX(r) establishes the one-to-one correspondence between affine maps of the set of density operators to the set of probability distributions on $R_{M}$ and the resolution of identities. In some cases, X(r) is called the positive operator valued measure (POVM).*


This formulation makes it possible to formulate the Born effect in quantum measurements without going through concrete physical quantities (observable).

### 3.2. Quantum Decision Operator

In quantum communication theory, the POVM is used as the decision operator $\left\{\Pi_{m}\right\}$ by following operational correspondence.

Let us assume the set $\rho_{m},m=\left\{1,2,3,\ldots ,M\right\}$ of quantum states. The decision result through the quantum measurement is described by a compact set of the decision operator $\Pi_{l}$, with l=1,2,3,…,M, such that(2)$$\begin{array}{ccc} P(l|m) & = & Tr\rho_{m}\Pi_{l},\;m,l=1,2,3,\ldots ,M \end{array}$$(3)$$\begin{array}{ccc} \sum_{l=1}^{M}\Pi_{l} & = & I,\;\Pi_{l}\geq 0\;\forall l \end{array}$$
where *I* is the identity operator.

**Remark** **2.**
*The decision operator does not just represent the probability of a quantum measurement process but the probability of error or detection by the receiver’s decision. In other words, it must be understood that it involves a decision by the observer [1].*


The above remark, from the origins of Helstrom’s formulation, is easier to understand if we interpret the quantum decision operator as the generalization of Wald’s decision function in a classical system [18].

In classical communication theory, the decision is applied to the given probability function of the variable of the received signal. When one applies the standard quantum measurement, the probability function is given, and the decision function is applied to its probability function as in classical detection theory. However, in the quantum case, the decision operator does not need the explicit probability function of the variable of the received signal. The decision operator directly outputs the probability value of the correctness or incorrectness of the decision without going through the probability function of the measurement process. As an effect of the above fact, the possibility that the discrimination capability of the quantum signal in the quantum measurement process will exceed the discrimination capability by the standard likelihood decision based on standard quantum measurement and its probability function arises. This is equivalent to violating communication-theoretic causality [19,20].

However, this can occur only in the discrete signal set. Its effect disappears in the case of continuous variables. That is, in quantum estimation theory, there is no such quantum advantage. These observations will be discussed in the subsequent sections.

### 3.3. Decision Operator Based on Entangled Measurement (or Collective Measurement)

As mentioned above, a quantum measurement process is considered a communication channel. In information theory, information is generally transmitted as a code word of length *n*. When extended to the *n*-th extension, quantum communication channels need to be treated differently from conventional extension channels.

Let us consider a finite alphabet $\left\{a_{l}\right\},l=\left(1,2,\ldots ,M\right)$ with quantum state $\rho_{l}$. The code word k is a combination of the alphabet $\left\{a_{l}\right\}$ with the length *n*. That is, the input code $k=(k_{1},k_{2},\ldots ,k_{i},\ldots ,k_{n})$, and the output is $j=(j_{1},j_{2},\ldots ,j_{i},\ldots ,j_{n})$, where *i* is the order of the sequence in the code, $0\leq k_{i}\leq M$, and $0\leq j_{i}\leq M$. The compound state $\rho_{k}=\left(\rho_{k_{1}}⊗\rho_{k_{2}}\cdots ⊗\rho_{k_{n}}\right)$ is in a tensor product Hilbert space $H^{⊗n}=H⊗\cdots ⊗H$. When the *n*-th extended signal state is used, the operation of treating a sequence of quantum states of length *n* as a single quantum state in the *n*-th extended Hilbert space is called collective or entangled measurement-based decoding. In this case, these structures provide unique performance in quantum Shannon information theory, which will be described in the next section. In cases of individual and entangled measurement, optimizing the decision operator that represents the decision process is an important issue.

## 4. Structure of Quantum Detection and Estimation Theory

### 4.1. Quantum Detection Theory

#### 4.1.1. Basic Formula

The appearance of quantum effects in the measurement process of signals and the result of the decision are simultaneously characterized by the quantum decision operator. Thus, the quantum Bayes rule is formulated without going through the likelihood ratio as follows:(4)$${\bar{P}}_{e}=\min_{\left\{\Pi \right\}}\left\{1-\sum_{m=1}^{M}\xi_{m}Tr\rho_{m}\Pi_{m}\right\}$$
where the a priori probability must be $\left(\xi_{m}>0,\;\forall m\right)$ for admissibility in decision theory. The necessary and sufficient conditions for $\left\{\Pi_{m}\right\}$ are given by Holevo [2] and Yuen [3].

**Theorem** **2**({Holevo,Yuen})**.**
*The necessary and sufficient condition for $\left\{\Pi_{m}\right\}$ on the quantum Bayes rule is given by*(5)$$\begin{array}{ccc} & & \Pi_{m}\left[\xi_{m}\rho_{m}-\xi_{l}\rho_{l}\right]\Pi_{l}=0,\;\forall l,m\\ & & \gamma -\xi_{l}\rho_{l}\geq 0,\;\forall l\\ & & \gamma =\sum_{l}\xi_{l}\rho_{l}\Pi_{l} \end{array}$$
*On the other hand, following classical detection theory, the quantum minimax rule for a signal system in which the functional with respect to $\left\{\xi_{m}\right\}$ is a convex function is defined as*

(6)
$${\bar{P}}_{e}=\max_{\left\{\xi \right\}}\min_{\left\{\Pi \right\}}\left\{1-\sum_{m=1}^{M}\xi_{m}Tr\rho_{m}\Pi_{m}\right\}$$

*and the necessary and sufficient conditions are as shown below [21].*


**Theorem** **3**({Hirota·Ikehara})**.**
*The necessary and sufficient condition for $\left\{\Pi_{m}\right\}$ on the quantum minimax rule is given by*(7)$$\begin{array}{ccc} & & Tr\Pi_{l}\rho_{l}=Tr\Pi_{m}\rho_{m},\;\forall l,m\\ & & \Pi_{m}\left[\xi_{m}\rho_{m}-\xi_{l}\rho_{l}\right]\Pi_{l}=0,\;\forall l,m\\ & & \gamma -\xi_{l}\rho_{l}\geq 0,\;\forall l\\ & & \gamma =\sum_{l}\xi_{l}\rho_{l}\Pi_{l} \end{array}$$
*For further formulae, please see the references.*


#### 4.1.2. Useful Analytical Issues

In general, it is very difficult to find the solutions of the above two quantum detection rules. However, when the signal system satisfies certain conditions or the decision operator can be set in advance, the optimal theory becomes very simple. Some examples of these are described below.

A set of quantum states that conforms to the following definition is particularly convenient.

**Definition** **4.**
*Let G be a group with an operation *∘*. The set of quantum state signals is called group covariant if there exist unitary operators $U_{k}\left(k\in G\right)$ such that*

(8)
$$U_{k}\left|\psi_{m}>=\right|\psi_{k\circ m}>,\forall m,k\in G$$


*It characterizes quantum states $\left\{|\psi_{m}>,m\in G\right\}$ [17].*


Let us explain the case of setting a decision operator in advance. Belavkin [22] introduced the following decision operator called square root measurement (SRM).

**Definition** **5.**
*The decision operator called a square root measurement (SRM) is given by*

(9)
$$\begin{array}{ccc} & & \Pi_{l}=\left|\mu_{l}><\mu_{l}\right|\\ & & |\mu_{l}>=\Gamma^{-1/2}|\alpha_{l}>\\ & & \Gamma =\sum_{m=1}^{M}\left|\alpha_{m}><\alpha_{m}\right| \end{array}$$

*where *Γ* is a Gram operator.*

*Based on the above formula, the general properties of the quantum Bayes rule and quantum minimax rule based on the above decision operator are given by Ban and Osaki [23,24].*


#### 4.1.3. Decision Operator Based on SRM of Entangled Measurement

In the case of the operation of mapping an individual decision to the quantum states corresponding to each slot of a code word, the decision operator and decision probability are given by(10)$$\begin{array}{ccc} \Pi_{j} & = & \Pi_{j_{1}}⊗\Pi_{j_{2}}\cdots ⊗\Pi_{j_{n}}\\ P(j|k) & = & P\left(j_{1}|k_{1}\right)\times P\left(j_{2}|k_{2}\right)\times \cdots \times P\left(j_{n}|k_{n}\right) \end{array}$$
where $P\left(j_{l}|k_{l}\right)=Tr\rho_{j_{l}}\Pi_{j_{l}}$, for l={1,2,…,n}. In this case, there is no correlation in the signal decision process.

On the other hand, one can define the decision operator based on the entangled measurement. Let us denote the quantum state of *n*-th extended quantum state as follows:(11)$$|\Psi_{k}>=|\psi_{k_{1}}>|\psi_{k_{2}}\ldots \left|\psi_{k_{l}}>\ldots \right|\psi_{k_{n}}>$$
where $k_{l}=\left\{1,2,\ldots ,M\right\},\forall l$. Then we can adopt the square root measurement as follows:(12)$$\begin{array}{ccc} & & \Pi_{j}=\left|\Phi_{j}><\Phi_{j}\right|,\\ & & |\Phi_{j}>=(\sum_{k}|\Psi_{k}><\Psi_{k}{\left|\right)}^{-1/2}|\Psi_{j}>\\ & & P\left(j|k\right)=Tr\rho_{k}\Pi_{j} \end{array}$$
where $\rho_{k}=\left|\Psi_{k}><\Psi_{k}\right|$. This is called decision operator based on entangled measurement. In this case, in addition to the quantum effects due to the decision action on each alphabet, quantum effects due to entanglement can be expected.

#### 4.1.4. Quantum Advantage in Detection

The performance of optical communications, which is designed using classical communication theory, is determined by the performance of existing receivers such as photon-counting, homodyne, and heterodyne receivers. If the performance of a communication system designed using quantum theory exceeds conventional performance, it is called quantum advantage or quantum gain. The quantum gain characteristics by the quantum decision operator are exemplified below. The concrete advantage of the decision operator based on entangled measurement is discussed in Section 5.

**Theorem** **4.**
*If the signal set $\left\{|\alpha_{m}>\right\}$ is covariant, the optimum decision operator is given by the decision operator based on the square root measurement. And the error probability of quantum Bayes and quantum minimax rules for M covariant signals can be given as follows:*

(13)
$$\begin{array}{ccc} {\bar{P}}_{e} & = & 1-\frac{1}{{\left(M\right)}^{2}}(\sum_{m=1}^{M}{\sqrt{\lambda }}_{m}{)}^{2}\\ \lambda_{m} & = & \sum_{k=1}^{M}<\alpha_{1}|\alpha_{k}>u^{-(k-1)m} \end{array}$$

*where u=exp[πi/M].*


Here we show the simplest example. Assuming that the signal system is binary PSK $\rho_{1}=|\alpha ><\alpha |,\rho_{2}=\left|-\alpha ><-\alpha \right|$ with $\xi_{1}=\xi_{2}=1/2$, the quantum solution is as follows [1]:(14)$$\begin{array}{ccc} {\bar{P}}_{e}^{opt} & = & 1-(Tr\rho_{1}\Pi_{1}^{opt}+Tr\rho_{2}\Pi_{2}^{opt})\\ & = & \frac{1}{2}\left[1-\sqrt{{1-|<\alpha |-\alpha >|}^{2}}\right]\\ & \ll & {\bar{P}}_{e}\left(Homodyne\right) \end{array}$$

This is called the Helstrom bound. The decision operator in this case is given as follows:(15)$$\Pi_{1}^{opt}=|\omega_{1}><\omega_{1}|,\;\Pi_{2}^{opt}=\left|\omega_{2}><\omega_{2}\right|$$
where(16)$$\begin{array}{ccc} |\omega_{1}> & = & \sqrt{\frac{1+\sqrt{1-\kappa^{2}}}{2(1-\kappa^{2})}}|\alpha >\\ & - & \sqrt{\frac{1-\sqrt{1-\kappa^{2}}}{2(1-\kappa^{2})}}|-\alpha >\\ |\omega_{2}> & = & \sqrt{\frac{1-\sqrt{1-\kappa^{2}}}{2(1-\kappa^{2})}}|\alpha >\\ & - & \sqrt{\frac{1+\sqrt{1-\kappa^{2}}}{2(1-\kappa^{2})}}|-\alpha > \end{array}$$
where κ=|<α|−α>|.

In the case of a homodyne receiver (classical solution), the measurement operator is(17)$$\Pi \left(x\right)=|x><x|,\;X_{c}\left|x>=x\right|x>,\;X_{c}=\frac{1}{2}\left(a+a^{†}\right)$$

The average error probability is given by the likelihood test based on $P\left(x|\alpha \right)=Tr\rho_{1}\Pi \left(x\right),P\left(x|-\alpha \right)=Tr\rho_{2}\Pi \left(x\right)$. It corresponds to the post-measurement procedure. The result does not show quantum advantage.

Separate analysis is required to determine what physical process the quantum optimal solution in Equation (Equation 16) corresponds to. An example is the Dolinar receiver [1].

### 4.2. Quantum Estimation Theory

Quantum estimation theory was also pioneered by Helstrom, Holevo, Personick, and Yuen. Then M.G.A. Paris et al. discussed many aspect of its application [25]. In this section, we show the formulation and discuss a certain problem.

#### 4.2.1. Formulation

When a physical signal parameter is continuous, a set of the quantum states can be treated as a quantum state system corresponding to a continuous signal. Optimization theory goes to quantum estimation from quantum detection theory. Therefore, the decision must deal with a continuous quantity {x}. So for the density operator ρ(x), the decision is defined as an infinitesimal operator.(18)$$\begin{array}{ccc} & & \left\{\Pi_{i}\right\}\to \left\{d\Pi \left(x\right)\right\}\\ & & P\left(x_{out}|x_{in}\right)=Tr\rho \left(x_{in}\right)d\Pi \left(x_{out}\right) \end{array}$$
where $x=x_{in}=x_{out}$ for the correct probability. Thus, quantum Bayes estimation can be formulated by replacing the discrete system with an infinitesimal decision operator. In this case, the optimum POVM corresponds to the measurement process of the observable as the parameter. That is, the optimum operator is given by the eigenstate of the operator of the physical observable. Thus, the estimation mechanism will use the maximum likelihood method, just like in classical methods, based on P(x)=Trρ(x)dΠ(x). Hence, it is easy to understand that it has no quantum advantage in the decision process.

Alternatively, one can adopt the minimum mean square error (MMSE) as the evaluation function. Personick formulated an MMSE estimator which is applicable to the linear filtering of random signal sequences [26]. On the other hand, the Cramér–Rao bound is formulated by Helstrom as follows [1]:

**Theorem** **5.**
*The estimation bound for a single parameter is given by the following formula:*

(19)
$$Var\hat{x}=Tr\rho {\left(\hat{x}-x\right)}^{2}\geq \frac{1}{Tr\rho \left(x\right)L_{S}^{2}}$$

*where $L_{S}$ is called the symmetric logarithm derivative and is given by*

(20)
$$\frac{\partial \rho (x)}{\partial x}=\frac{1}{2}\left[\rho \left(x\right)L_{S}+L_{S}\rho \left(x\right)\right]$$


*The equality holds in the following case:*

(21)
$$L_{S}=k\left(x\right)\left(\hat{x}-x\right)$$

*where $\hat{x}$ is the estimate, k(x) is a function of x, and the estimate operator $\hat{x}$ is given by the operator corresponding to the parameter.*


This means that the optimum measurement is the standard quantum measurement and that there is no quantum advantage in the measurement process itself. In addition, there is still a certain difficulty, as discussed in the following.

#### 4.2.2. Example for Coherent-State Signal

In optical communications and optical radar, signals generally have complex amplitudes (quadrature amplitude):(22)$$\alpha^{s}=x_{c}^{s}+ix_{s}^{s}=|\alpha^{s}\left|\cos\theta +i\right|\alpha^{s}|\sin\theta =\sqrt{N_{s}}e^{i\theta }$$

Its quantum counterpart is the photon annihilation operator $a=X_{c}+iX_{s}$, where(23)$$X_{c}=\frac{1}{2}\left(a+a^{†}\right),\;X_{s}=\frac{1}{2i}\left(a-a^{†}\right)$$

Here $X_{c}$ and $X_{s}$ are non-commutative observables. The problem is how to formulate the simultaneous estimation of such non-commutative observables.

(a) Single-parameter estimation

In general, the density operator of an optical field is given as follows:(24)$$\rho =\frac{1}{\pi N}\int \exp\left\{-\frac{|\alpha -\alpha^{s}{|}^{2}}{N}\right\}\left|\alpha ><\alpha \right|d^{2}\alpha$$
where *N* is background noise energy and $\alpha^{s}$ is the signal. When we consider the estimation of the single parameter $x_{c}^{s}$ of $\alpha^{s}=x_{c}^{s}+ix_{s}^{s}$, the solution is given as follows:(25)$$\begin{array}{ccc} \frac{\partial \rho }{\partial x_{c}^{s}} & = & \frac{1}{2}\left(\rho L_{S}+L_{S}\rho \right) \end{array}$$(26)$$\begin{array}{ccc} L_{S} & = & \frac{2}{N+1/2}\left(\frac{a+a^{†}}{2}-x_{c}^{s}\right) \end{array}$$

As a result, we have(27)$$\hat{x}=X_{c}=\frac{1}{2}\left(a+a^{†}\right),\;Var{\hat{x}}_{c}=\frac{1}{2}N+\frac{1}{4}$$
where 1/4 is the effect of quantum noise. The estimate operator $\hat{x}$ indicates a homodyne receiver. Thus, it has no quantum advantage.

(b) Non-commutative parameter estimation

Let us consider the simultaneous estimation of the non-commutative observables $X_{c}$ and $X_{s}$. This is important for the evaluation of optical phase estimation such as(28)$$\theta =\tan^{-1}\frac{x_{s}}{x_{c}}$$

In this case, Theorem 5 does not provide the correct solution for the design of a communication system. The quantum optimum measurement for non-commutative quadrature amplitude can be given by the Yuen–Lax quantum Cramér–Rao bound [27].

**Theorem** **6.**
*The estimation bound for complex amplitudes is given by the following formula:*

(29)
$$Var\left(\hat{\alpha }\right)\geq \frac{1}{Tr\rho L_{R}L_{R}^{†}}$$

*where the right logarithm derivative $L_{R}$ is defined by*

(30)
$$\frac{\partial \rho }{\partial \alpha^{s}}=L_{R}^{†}\rho ,\;L_{R}=k\left(\alpha^{s}\right)\left(a-\alpha^{s}\right)$$

*where a is a photon annihilation operator and it corresponds to a heterodyne measurement. Then the bound is given as follows:*

(31)
$$Var\hat{\alpha }=N+1$$

*where 1=1/2+1/2 is the simultaneous measurement effect of the non-commutable observables.*


If one adopts the separate measurement of the quadrature amplitude by two balanced homodyne receivers, from Theorem 1, the coherent state maintains the coherent state at the output of the half mirror. Here each measurement may adopt Theorem 5.(32)$$\begin{array}{ccc} \frac{\partial \rho (x_{c}^{s})}{\partial x_{c}^{s}} & = & \frac{1}{2}\left[\rho \left(x_{c}^{s}\right)L_{S}+L_{S}\rho \left(x_{c}^{s}\right)\right] \end{array}$$(33)$$\begin{array}{ccc} \frac{\partial \rho (x_{s}^{s})}{\partial x_{s}^{s}} & = & \frac{1}{2}\left[\rho \left(x_{s}^{s}\right)L_{S}+L_{S}\rho \left(x_{s}^{s}\right)\right] \end{array}$$

Each bound is given by(34)$$Var{\hat{x}}_{c}=\frac{1}{2}N+\frac{1}{4},\;Var{\hat{x}}_{s}=\frac{1}{2}N+\frac{1}{4}$$

However, one should not ignore the fact that the signal energy is half at each branch. Thus, the signal-to-noise ratio for the quantum part is(35)$$\begin{array}{ccc} SNR(x_{c}) & = & \frac{N_{s}/2}{1/4}=\frac{N_{s}}{1/2},\\ SNR(x_{s}) & = & \frac{N_{s}/2}{1/4}=\frac{N_{s}}{1/2}, \end{array}$$

This means that a set of two balanced homodyne receiver systems is equivalent to a heterodyne receiver for the coherent-state signal. Theorem 6 includes the above effect under fixed energy conditions.

Now consider the case where the signals are in a state other than coherent state. In separate measurement of non-commutative parameters, the quantum state is disturbed by a loss of half mirror. Therefore, it is not accurate to evaluate communication performance based on the conventional quantum Cramér–Rao bound only.

#### 4.2.3. Application of Lie Algebra for Non-Commutative Parameters

We have been studying the theory of quantum state control and simultaneous measurement of non-commutative quantities based on Lie algebra. Here we introduce it and its application to the formulation of estimation theory. First, we define the formulation as follows.

**Definition** **6.**
*Algorithm:*

*Step 1: Derive the estimation operators for simultaneous estimation of non-commutative quantities based on the symmetric logarithmic differential operator formulae in Equations (32) and (33).*

*Step 2: Construct a decision operator that expresses simultaneous measurement of non-commuting quantities based on the minimum uncertain state for the two operators obtained.*



(a) Construction of decision operator

**Definition** **7.**
*An algebra is called a Lie algebra when its multiplication satisfies the following two conditions:*

(36)
$$\begin{array}{ccc} & & [A,A]=0,\\ & & [[A,B],C]+[[B,C],A]+[[C,A],B]=0 \end{array}$$

*where [] is the commutation relation.*


An example of its application in quantum mechanics is the Heisenberg–Weyl group: $W_{1}$. The unitary representation of the subgroup of $W_{1}$ is(37)$$T\left(g\right)=\exp\left\{is\right\}\exp\left\{\alpha a^{†}-\alpha^{*}a\right\}$$
where g is the element of Lie algebra.

**Definition** **8.**
*For any element |ψ> of a Hilbert space, the quantum state constructed by the formula*

(38)
$$T\left(g\right)|\psi >=|\alpha_{\psi }{>}_{G}$$

*is called the generalized coherent state.*


Here let us consider non-compact groups in Lie algebras. With respect to Hermite forms, a set that constitutes linear isometric groups is a non-compact group. SU(1,1) is a typical example.

The Lie algebra of SU(1,1) has the following elements: $K_{1},K_{2},K_{0}$. Their commutation relations are(39)$$\begin{array}{ccc} & & \left[K_{1},K_{2}\right]=-iK_{0},\left[K_{2},K_{0}\right]=iK_{1}\\ & & \left[K_{0},K_{1}\right]=iK_{2} \end{array}$$

Then we define $K_{\pm }=K_{1}\pm iK_{2}$. The unitary representation of SU(1,1) is given by(40)$$T^{\tau }\left(g\right)=\exp\left\{\zeta K_{+}-\zeta^{*}K_{-}\right\}$$
where τ is given by the property of the Casimir operator. The following state is also the generalized coherent state:(41)$$T^{\tau }\left(g\right)\left|\psi >=\right|\alpha_{\tau }{>}_{G}$$
where $|\alpha_{\tau }>$ is called a base state. When the above state satisfies the minimum uncertainty relation for two observables, it is called the ideal generalized coherent state.

Here, for the quadrature amplitude, the elements of SU(1,1) are as follows:(42)$$\begin{array}{ccc} & & K_{+}=\frac{1}{2}{a^{†}}^{2},K_{-}=\frac{1}{2}a^{2},\\ & & K_{0}=\frac{1}{4}\left(aa^{†}+a^{†}a\right) \end{array}$$

Then we have(43)$$\begin{array}{ccc} & & T^{\tau }\left(g\right)=\exp\left\{\frac{z}{2}\left(a^{2}-{a^{†}}^{2}\right)\right\} \end{array}$$(44)$$\begin{array}{ccc} & & T^{\tau }\left(g\right)a{T^{\tau }}^{†}\left(g\right)=\mu a+\nu a^{†}\equiv b\\ & & |\alpha_{\tau }>={\left|0>,\;|\mu \right|}^{2}-{\left|\nu \right|}^{2}=1 \end{array}$$

Then the following state is the ideal generalized coherent state:(45)$$\begin{array}{ccc} & & b|\beta ,\mu ,\nu >=\beta |\beta ,\mu ,\nu > \end{array}$$(46)$$\begin{array}{ccc} & & T^{\tau }\left(g\right)\left|0>=\right|0,\mu ,\nu > \end{array}$$
where $\beta =\mu \alpha +\nu \alpha^{*}={\bar{x}}_{c}^{G}+i{\bar{x}}_{s}^{G}$. Equations (45) and (46) give the minimum uncertainty state for $x_{c}$ and $x_{s}$.

Let us define the decision operator as step 1:(47)$$d\Pi (x_{c},x_{s})=|\beta ,\mu ,\nu ><\beta ,\mu ,\nu |$$

This corresponds to the generalized heterodyne receiver [28]. Let us show how to use the above formula in the following.

(b) Estimation bound

Using these formulae, the modified quantum Cramér–Rao bound for the simultaneous estimation of non-commutative parameters of a coherent-state signal without background noise is given by step 2 as follows [28]:(48)$$\begin{array}{ccc} & & Var{\hat{x}}_{c}=\frac{1}{4}+\frac{1}{4}{\left|\mu \pm \nu \right|}^{2}\\ & & Var{\hat{x}}_{s}=\frac{1}{4}+\frac{1}{4}{\left|\mu \mp \nu \right|}^{2} \end{array}$$

Then the total minimum bound is(49)$$\min_{\mu ,\nu }\left(Var{\hat{x}}_{c}+Var{\hat{x}}_{c}\right)=1,\;\mu =1,\nu =0$$

This is equivalent to the Yuen–Lax bound.

#### 4.2.4. Importance of Signal-to-Noise Ratio in Estimation

In general, one should keep the following in mind: The performance of the estimation cannot be evaluated only by the estimation bound. Following communication theory, one needs the concept of the signal-to-noise ratio, which means the operational meaning in the technology.

Let us show an example of the evaluation for a continuous parameter. The squeezed state reduces the quantum noise of one of the quadrature amplitudes, and the purpose is to utilize the signal with that low noise. However, we have the following formulation.

**Theorem** **7**({Yuen} [29])**.**
*We assume that $\rho =|\beta ,\mu_{s},\nu_{s}><\beta ,\mu_{s},\nu_{s}|$ as a signal state. Let us define the SNR of this state as follows:*(50)$$SNR=\frac{{\left(Tr\rho X_{c}\right)}^{2}}{Tr\rho {\left(X_{c}-<X_{c}>\right)}^{2}}$$
*Under the constraint of the signal energy $Tr\rho a^{†}a\leq N_{s}$, the maximum SNR is given by*

(51)
$$SNR_{max}=4N_{s}\left(N_{s}+1\right)$$

*where*

(52)
$$\mu_{s}=\frac{N_{s}+1}{\sqrt{2N_{s}+1}},\;\nu_{s}=\frac{N_{s}}{\sqrt{2N_{s}+1}}$$


*The estimation bound is given by*

(53)
$$Var{\hat{x}}_{c}=\frac{1}{4}{\left|\mu_{s}-\nu_{s}\right|}^{2}=\frac{1}{4}\times \frac{1}{2N_{s}+1}$$


*When the state is a coherent state, $SNR=4N_{s}$.*


As mentioned above, low noise does not mean good performance. The SNR is the final evaluation for communication technology.

When there is an energy loss ϵ in the channel, the SNR is(54)$$SNR=\frac{4\epsilon N_{s}\left(N_{s}+1\right)}{\epsilon +\left(1-\epsilon \right)\left(2N_{s}+1\right)}$$

The estimation bound is(55)$$Var{\hat{x}}_{c}=\frac{\epsilon }{4}\times \frac{1}{2N_{s}+1}+\frac{1-\epsilon }{4}$$

When the loss is large, the above SNR becomes $SNR=2N_{s}$, which is worse than the coherent state.

Since evaluations for problems in technology require operational meaning, we emphasize the points to note in “quantum communication theory and quantum mathematical statistics” (Figure 3).

## 5. Quantum Shannon Information Transmission Theory

Shannon theory is a design theory for transmitting signals through noisy communication channels with as few errors as possible. Shannon defined mutual information as the evaluation function, and its maximum value is called the channel capacity. Its operational meaning is the maximum rate (a kind of efficiency) at which errors can be minimized.

In this section, we explain the foundation of design theory for systems in which communication signals are transmitted by optical signals of quantum nature. From Theorem 1, it is necessary to assume a coherent state for quantum design to provide advantages over existing communication capabilities in real-world environments.

### 5.1. Channel Capacity

#### 5.1.1. Finite, Discrete Alphabet System

Shannon mutual information in the quantum measurement channel is written as follows: (56)$$J_{1}\left(\xi ,\Pi \right)=\sum_{i=1}^{M}\sum_{j=1}^{M^{\prime }}\xi_{i}P\left(j|i\right)\ln\left[\frac{P(j|i)}{\sum_{k}\xi_{k}P\left(j|k\right)}\right]$$
where $P\left(j|i\right)=Tr\rho_{i}\Pi_{j}$. The theory of maximization of mutual information is twinned with the above quantum decision theory, and it is denoted as follows: (57)$$C_{1}=\underset{\xi ,\Pi }{\sup}J_{1}\left(\xi ,\Pi \right)$$

Thus, it has the same theoretical structure as the optimal theory for detection theory. As a result, the basic result was derived by Holevo [2] as shown below.

**Theorem** **8**({Holevo})**.**
*The necessary condition for maximum mutual information with respect to the decision operators for a simple set of states is given as follows:*(58)$$\begin{array}{ccc} & & P\left(j|i\right)=Tr\rho_{i}\Pi_{j}\\ & & F_{j}=\sum_{i}\xi_{i}\rho_{i}\log\left\{\frac{P(j|i)}{\sum_{l}\xi_{l}P\left(j|l\right)}\right\}\\ & & \Pi_{j}\left[F_{j}-F_{i}\right]\Pi_{i}=0,\forall i,j \end{array}$$
*where the solution is called accessible information.*

On the other hand, there is a problem of Davies in the case of the optimization of mutual information [30]. A partial solution to this is given by [31] under Jozsa’s guidance.

We consider the concrete property of the above. Several properties of mutual information were discussed in the serial papers by M. Osaki [32]. As in signal detection theory, the mutual information obtained by the optimal decision operator is greater than that obtained by a classical measurement process. The most useful results are as follows.

**Remark** **3.**
*When the signal states are group covariant, the quantum Bayes and minimax decision operators satisfy the above necessary condition for mutual information. In a real environment with large losses or background noise, no quantum state exists that can exceed the mutual information provided by communication based on coherent states.*


Now we still have a difficult problem that is proof of sufficiency. According to Osaki’s analysis, the quantum minimax decision operator may provide the maximum mutual information in the practical region such as ${\left|\alpha \right|}^{2}\gg 1$ in the set of coherent states.

On the other hand, let us consider the *n*-th extension of the alphabet and adopt the decision operator based on entangled measurement. Then the following conjecture is imposed:(59)$$C_{1}\leq C_{2}\leq \cdots \leq C_{n}\cdots \leq C_{\infty }=\lim_{n\to \infty }\frac{1}{n}C_{n}$$

Hausladen et al. [4] proved it for the pure-state channel, and Holevo, Schumacher, and Westmoreland proved it for the general states [5,6].

**Theorem** **9**({Holevo·Schumacher·Westmoreland})**.**
*The upper bound of maximum mutual information and the capacity are given by Holevo information as follows:*(60)$$\begin{array}{ccc} & & C_{1}\leq S\left(\rho_{T}\right)-\sum_{i=1}^{M}\xi_{i}S\left(\rho_{i}\right)=I_{H}\\ & & where\;\rho_{T}=\sum_{k=i}^{M}\xi_{i}\rho_{i}, \end{array}$$(61)$$\begin{array}{ccc} & & Then\;C_{\infty }=C_{Holevo}=\max_{\xi }I_{H} \end{array}$$
*where S(ρ)=−Trρlogρ is the von Neumann entropy.*

The above theorem is the channel capacity formula for a quantum Shannon information transmission system.

**Remark** **4.**
*The maximum absolute value of Holevo·Schumacher·Westmoreland (HSW) capacity in a real environment is given by the coherent state.*


The numerical properties of HSW capacity for coherent-state signals have been analyzed by our group [33,34].

#### 5.1.2. Infinite Alphabet System (Continuous)

After the analysis for discrete alphabet systems, the analysis was extended to analog signals with constrained power inputs, among which are channels with additive quantum Gaussian noise. This is the issue of the infinite alphabet scheme. Here we denote a simple result on the final capacity formula of quantum Gaussian channels of free space. According to the quantum Shannon theory established by Holevo and others, the capacity formula of optical quantum communication for free space lossy Gaussian channels is discussed in [35,36,37].

**Theorem** **10**({Holevo·Sohma·Hirota})**.**
*The capacity formula of the single-mode quantum lossy Gaussian noise channel is given as follows [36]:*(62)$$\begin{array}{ccc} C_{Holevo} & = & g\left(N_{s}+N_{Th}\right)-g\left(N_{Th}\right)\\ & = & \left(N_{s}+N_{Th}+1\right)\log\left(N_{s}+N_{Th}+1\right)\\ & & -\left(N_{s}+N_{Th}\right)\log\left(N_{s}+N_{Th}\right)\\ & & -\left(N_{Th}+1\right)\log\left(N_{Th}+1\right)\\ & & +N_{Th}\log N_{Th} \end{array}$$
*where the above is given by the coherent state; g(x)=(x+1)log(x+1)−xlogx, $N_{s}$ and $N_{Th}$ are average photon numbers of the signal and additive noise at the receiver; and *1* of $\left(1+N_{Th}\right)$ means the quantum noise.*

The above formula is, in general, greater than the Shannon classical capacity.(63)$$C_{Shannon}=\log\left(1+\frac{N_{s}}{1+N_{Th}}\right)$$
where the above formula is given by a heterodyne receiver. Thus, the main parameter of capacity in Shannon theory for continuous alphabets is the SNR.

#### 5.1.3. Quantum Advantage in Capacity

Let us discuss quantum advantage. In the transmission system of Shannon information, the quantum advantage for a discrete alphabet is given by(64)$$C\left(classical\right)\leq C_{1}\leq C_{H}$$
where C(classical) means the capacity for a digital optical fiber communication system designed by conventional communication theory. The above relation of quantum advantages results from the error mitigation effect in the quantum decision mechanism and coding designed by quantum detection theory.

On the other hand, the quantum advantage in the case of infinite alphabets (continuous) is given by(65)$$C_{Holevo}-C_{Shannon}\geq 0$$

When the background noise is very large, the quantum advantage disappears.

#### 5.1.4. Implementation Problem

It is important to discuss technologies that will actually realize quantum gain in terms of capacity. The quantum gain for $C_{1}$ comes from the effect of the decision operator of a single shot. The quantum gain of $C_{H}$ comes from both the decision operator based on entangled measurement and coding scheme. Below are some examples of discrete and continuous systems.

(a) Finite, discrete case

The problem, in this case, is an issue of superadditivity of the capacity $C_{n}$ with respect to *n*. There are two elements to achieving superadditivity. One is code construction consisting of a sequence of quantum states. The second is a construction of decision operator and its generalization based on entangled measurement; see Equation (Equation 12). Many pioneering analyses of the former have already been published, but the latter remains a very difficult problem. Several challenging attempts based on measurement effect have been made [38,39,40,41], and some analytical examples based on coding and measurement are given in [42,43].

(b) Continuous case

Generally, quantum gain disappears when the background noise is large or the signal energy is large. Therefore, we discussed how to achieve this when there is no background noise and the received signal is very weak from an ultra-long distance. In this case, it is known that the capacity can be attained by discretization [44].

### 5.2. Quantum Reliability Function and Quantum Cut-Off Rate

#### 5.2.1. Reliability Function

The operational meaning of Shannon mutual information is the efficiency of coding and decoding in a noisy channel. The unit is bits/symbol, and it is not the actual amount of information. Reliability functions are introduced to clarify the operational meaning of Shannon mutual information and channel capacity. Here, we show their quantum equivalents. In general, the reliability function is defined as follows: (66)$$E_{Q^{\prime }}\left(R\right)=\lim_{n\to \infty }\sup\frac{1}{n}\ln\frac{1}{P_{e}}$$

The upper bound of the average error probability of a code of length *n* is as follows: (67)$$P_{e}\leq e^{-nE_{Q^{\prime }}\left(R\right)}$$

We will avoid mathematical details and discuss this in a simplified form.

**Theorem** **11**({Burnashev·Holevo} [8,9])**.**
*The upper bound on the average error probability of a pure-state code for a quantum channel is given by*(68)$$\begin{array}{ccc} P_{e} & \leq & 2\exp\{-\max_{\xi }\max_{0\leq s\leq 1}n\left[\mu_{Q}\left(\rho_{\xi },s\right)-sR\right]\}\\ & \equiv & exp[-nE_{Q}\left(R\right)] \end{array}$$
*where*
(69)$$\begin{array}{ccc} E_{Q}\left(R\right) & = & \max_{\xi }\max_{0\leq s\leq 1}n\left[\mu_{Q}\left(\rho_{\xi },s\right)-sR\right] \end{array}$$
(70)$$\begin{array}{ccc} \mu_{Q}\left(\rho_{\xi },s\right) & = & -\ln[Tr\rho_{\xi }^{1+s}],\\ \rho_{\xi } & = & \sum_{j=1}^{M}\xi_{j}\left|\psi_{j}><\psi_{j}\right| \end{array}$$
*where M is the number of symbols.*

Now the maximization problem becomes(71)$$\frac{\partial }{\partial s}\left[\mu \left(s,\xi \right)-\mathrm{sR}\right]=\frac{\partial \left[\mu \left(s,\xi \right)\right]}{\partial s}-R=0$$(72)$$\frac{\partial \mu (s,\xi )}{\partial s}=\frac{-\mathrm{Tr}\rho_{\xi }^{1+s}\ln\rho_{\xi }}{\mathrm{Tr}\rho_{\xi }^{1+s}}=\frac{-\sum \lambda_{j}^{1+s}\ln\lambda_{j}}{\sum \lambda_{j}^{1+s}}$$
where λ is an eigenvalue of $\rho_{\xi }$ and it is given by the eigenvalue of the following matrix:(73)$$\left[\begin{array}{ccc} \sqrt{\xi_{1}\xi_{1}}\langle \psi_{1}|\psi_{1}\rangle & \ldots & \sqrt{\xi_{1}\xi_{M}}\langle \psi_{1}|\psi_{M}\rangle \\ \vdots & \ddots & \vdots \\ \sqrt{\xi_{M}\xi_{1}}\langle \psi_{M}|\psi_{1}\rangle & \ldots & \sqrt{\xi_{M}\xi_{M}}\langle \psi_{M}|\psi_{M}\rangle \end{array}\right]$$

The above formula assumes the decision operator based on entangled measurement.

If quantum measurements are individual measurements, the reliability function is given by(74)$$E_{semi}=-sR-\ln\sum_{j=1}^{M^{\prime }}{\left(\sum_{i=1}^{M}\xi_{i}{P\left(j|i\right)}^{1/1+s}\right)}^{1+s}$$
where $P\left(j|i\right)=Tr\rho_{i}\Pi_{j}$ and the optimum {Π} is given by the minimum error probability conditions [10,11,12]. So this is called semi-quantum reliability function. Ban and Kurokawa clarified the difference between the two definitions [13,14].

#### 5.2.2. Quantum Cut-Off Rate

The quantum cut-off rate is defined from formulation of the quantum reliability function of Burnashev · Holevo in the paper of Figure 4. It is given by [13,14]

**Definition** **9.**
*The quantum cut-off rate is defined as follows.*

(75)
$$\begin{array}{ccc} R_{Q} & \equiv & \max_{\left\{\xi_{i}\right\}}\left[\mu \left({\rho_{\xi }}_{i},s=1\right)\right]\\ & = & \max_{\left\{\xi_{i}\right\}}\left[-\ln\sum_{i=1}^{M}\sum_{j=1}^{M}\xi_{i}\xi_{j}{\left|\langle \psi_{i}|\psi_{j}\rangle \right|}^{2}\right]\\ & = & -\ln\left[\min_{\left\{\xi_{i}\right\}}\sum_{i=1}^{M}\sum_{j=1}^{M}{\left(\Gamma \right)}_{ij}\xi_{i}\xi_{j}\right] \end{array}$$

*where ${\left(\Gamma \right)}_{ij}=|<\psi_{i}{\left|\psi_{j}>\right|}^{2}$.*


When the optimization with respect to ξ is performed, we have the following simple form.

**Theorem** **12**({Ban·Kurokawa·Hirota})**.**
*The quantum cut-off rate is given by*(76)$$R_{Q}=-\ln\left[\sum_{i=1}^{M}\sum_{j=1}^{M}{\left(\Gamma^{-1}\right)}_{ij}\right].$$
*under the condition for the optimum values of ξ as follows:*
(77)$${\tilde{\xi }}_{i}=\frac{\sum_{j=1}^{M}{\left(\Gamma^{-1}\right)}_{ij}}{\sum_{i=1}^{M}\sum_{j=1}^{M}{\left(\Gamma^{-1}\right)}_{ij}}>0\;\forall i$$
*When the a priori probability is uniform, it becomes*

(78)
$$\begin{array}{ccc} R_{Q} & = & \ln\left[\frac{M}{\sum_{j=1}^{s}{\left|\langle \psi_{i}|\psi_{j}\rangle \right|}^{2}}\right]\\ & = & \ln\left[\frac{M}{\langle \psi |\hat{\Phi }|\psi \rangle }\right]. \end{array}$$

*where $\hat{\Phi }=\sum_{i=1}^{M}\left|\psi_{i}><\psi_{i}\right|$.*

*Thus, the relation between the reliability function and the cut-off rate is given by*

(79)
$$P_{e}\leq 2\exp\left[-nE_{Q}\left(R\right)\right]\leq 2\exp\left[{-n(R}_{Q}-R)\right].$$



Here let us introduce the definition by Helstrom based on individual measurement.

**Definition** **10.**
*Here a cut-off rate based on quantum individual measurement is defined as follows:*

(80)
$$R_{semi}=\max_{\left\{\xi_{i}\right\}}\max_{\left\{{\hat{\Pi }}_{i}\right\}}\left(-\ln\sum_{j=1}^{M^{\prime }}{\left[\sum_{i=1}^{M}\xi_{i}\sqrt{\left[\mathrm{Tr}{\hat{\rho }}_{i}{\hat{\Pi }}_{j}\right]}\right]}^{2}\right)$$


*And the upper bound of the above formula is given by*

(81)
$${\tilde{R}}_{semi}=\max_{\left\{\xi_{i}\right\}}\left[-\ln\sum_{i=1}^{M}\sum_{j=1}^{M}{\left(G\right)}_{ij}\xi_{i}\xi_{j}\right]$$

*where $G_{i,j}=|<\psi_{i}\left|\psi_{j}>\right|$. As a result, we have*

(82)
$$R_{semi}\leq R_{Q}\leq E_{Q}$$


*The example will be given in the next section.*


In the above, we introduced the special case of the pure state. For the general case, refer to Holevo’s paper [9].

## 6. Examples of Reliability Function and Cut-Off Rate

### 6.1. Finite, Discrete Alphabet System

Let us show some examples of the reliability function and cut-off rate for the finite alphabet.

#### 6.1.1. Analytical Method

The signal system is set to 3-ary PSK, which is composed of coherent states that enable ultra-high speeds over long distances.(83)$$|\psi_{1}>=\left|\alpha >,\;\right|\psi_{2}>=|\alpha e^{i\frac{2}{3}\pi }>,\;\left|\psi_{3}>=\right|\alpha e^{-i\frac{2}{3}\pi }>$$

In this case, the modified Gram matrix is as follows:(84)$$\frac{1}{3}\left[\begin{array}{ccc} 1 & K_{c}+iK_{s} & K_{c}-iK_{s}\\ K_{c}-iK_{s} & 1 & K_{c}+iK_{s}\\ K_{c}+iK_{s} & K_{c}-iK_{s} & 1 \end{array}\right],$$

The eigenvalues of the above are(85)$$\begin{array}{ccc} \lambda_{1} & = & \frac{1+2K_{c}}{3},\\ \lambda_{2} & = & \frac{1-K_{c}-\sqrt{3}K_{s}}{3},\\ \lambda_{3} & = & \frac{1-K_{c}+\sqrt{3}K_{s}}{3} \end{array}$$
where(86)$$\begin{array}{ccc} K_{c} & = & \exp\left[-\frac{3N_{s}}{2}\right]\cos\left[\frac{\sqrt{3}N_{s}}{2}\right],\\ K_{s} & = & \exp\left[-\frac{3N_{s}}{2}\right]\sin\left[\frac{\sqrt{3}N_{s}}{2}\right] \end{array}$$
where $N_{s}$ is the average photon number per pulse at the receiver. From these, we can derive the quantum reliability function and quantum cut-off.

On the other hand, if we assume individual measurements, we will adopt individual optimal decisions. In that case, the elements of the communication channel will be as follows:(87)$$\begin{array}{ccc} P(1|1) & = & P(2|2)=P(3|3)\\ & = & \frac{1}{9}\{3+2\sqrt{\beta_{1}\beta_{2}}+\\ & & 2\sqrt{\beta_{2}\beta_{3}}+2\sqrt{\beta_{3}\beta_{1}}\},\\ P(1|2) & = & P(2|3)=P(3|1),\\ P(1|3) & = & P(2|1)=P(3|2),\\ & = & \frac{1}{9}\{3-\sqrt{\beta_{1}\beta_{2}}-\\ & & \sqrt{\beta_{2}\beta_{3}}-\sqrt{\beta_{3}\beta_{1}}\} \end{array}$$
where(88)$$\begin{array}{ccc} \beta_{1} & = & 1+2K_{c},\\ \beta_{2} & = & 1-K_{c}-\sqrt{3}K_{s},\\ \beta_{3} & = & 1-K_{c}+\sqrt{3}K_{s} \end{array}$$

Figure 5 shows a numerical example. The communication system we set up here is intended for deep space communication, where the transmitted signal is sufficiently large and the transmission attenuation is extremely large, resulting in an extremely small received signal.

#### 6.1.2. Quantum Advantage of Decision Operator Based on Entangled Measurement

The role of quantum reliability function theory is to show how small the error rate of the transmitted code can be when the transmission rate of code is set. In the design theory of conventional optical communications, classical devices such as heterodynes and energy detectors are used as receivers, and reliability functions are evaluated using classical theory. The question is what benefits the quantum theory introduced here will bring to conventional optical communications. In quantum theory, quantum effects appear in quantum measurements, i.e., at the receiver. From the above theory, individual measurements only reduce the effect of quantum noise on individual signals. On the other hand, entangled measurements can take into account the quantum interference effect among signals due to measurement in addition to the individual quantum effects, thereby further reducing the quantum noise effect and increasing the reliability function and cut-off rate. As a result, when the code error rate is fixed (depending on the application), the code length required to achieve the same error rate can be significantly shortened. See Figure 6.

### 6.2. Infinite Alphabet System (Continuous)

#### 6.2.1. Reliability Function

The case of continuous systems with an infinite alphabet was introduced in the paper of Figure 7 [46]. This discussion is extremely important because it can clarify how quantum gain appears, i.e., in comparison with discrete systems.

Let us take the input alphabet *A*, an arbitrary Borel subset in a finite-dimensional Euclidean space. The input is described by an a priori probability ξ(x) on *A*. The energy constraint is posed as follows:(89)$$\int_{A}{\left(x-<x>\right)}^{2}\xi \left(x\right)dx\leq N_{s}$$

We adopt the product Hilbert space $H^{⊗n}=H⊗H⊗\cdots ⊗H$ with the input alphabet ^*n*^ consisting of code words $W=(x_{1},x_{2},\ldots ,x_{n})$ of length *n*, and the density operator(90)$$\rho_{W}=\rho \left(x_{1}\right)⊗\ldots \rho \left(x_{2}\right)\cdots ⊗\rho \left(x_{n}\right)$$

We define the set of code of size *M* that is a sequence {W(1),W(2),…,W(M)}. $\left\{\Pi_{j}^{c}\right\}$ is a family of decision operators in $H^{⊗n}$ based on entangled measurement, satisfying $\sum_{j=1}^{M}\Pi_{j}^{c}=I$. The error probability for a code word is $P\left(k|j\right)=Tr\rho_{W(j)}\Pi_{k}^{c}$, where j,k=1,2,…,M.

Let us consider the speed of the exponential decay of the error probability when n→∞ and $M=e^{nR}$ below the following capacity: (91)$$C_{H}=\max_{P(x)}[\hat{H}(\Lambda \left[\rho_{T}\left(\xi \left(x\right)\right)\right]-\int \hat{H}\left(\Lambda \left[\rho \left(x\right)\right]\right)\xi \left(x\right)dx$$

Following the Shannon idea of random coding, let us consider the random ensemble of *M* code words of length *n*. We have the following theorem [46].

**Theorem** **13**({Holevo})**.**
*Let us define $P_{e}\left(M=e^{nR},n\right)$ as the minimum error probability with respect to coding and the decision operator. Here we introduce the reliability function as follows:*(92)$$E\left(R\right)=\lim_{n\to \infty }\sup\frac{1}{n}\log\frac{1}{P_{e}\left(e^{nR},n\right)}$$
*The lower bound of the reliability function is given by*

(93)
$$E\left(R\right)\geq \max\{E_{Q_{r}}\left(R\right),E_{Q_{ex}}\left(R\right)\}$$

*where $Q_{r}$ and $Q_{ex}$ mean the random coding bound and expurgated bound, respectively. They are*

(94)
$$\begin{array}{ccc} E_{Q_{r}}\left(R\right) & = & \max_{0\leq s\leq 1}\left(\max_{0\leq p}\max_{\xi }\mu \left(\xi ,s,p\right)-sR\right) \end{array}$$


(95)
$$\begin{array}{ccc} E_{Q_{ex}}\left(R\right) & = & \max_{1\leq s}\left(\max_{0\leq p}\max_{\xi }\tilde{\mu }\left(\xi ,s,p\right)-sR\right) \end{array}$$

*where*

(96)
$$\begin{array}{ccc} \mu (\xi ,s,p) & = & -\log Tr{\left(\int e^{p[f(x)-E]}\rho \left(x\right)\xi \left(x\right)\right)}^{1+s}\\ \tilde{\mu }\left(\xi ,s,p\right) & = & -s\log\int \int e^{p[f(x)+f(y)-2E]}\\ & \times & {\left(Tr\sqrt{\rho (x)}\sqrt{\rho (y)}\right)}^{1/s}\xi \left(x\right)\xi \left(y\right) \end{array}$$

*where f is a function satisfying the condition for the central limit theorem for a random variable $f(x_{k})$*

(97)
$$\int f{\left(x\right)}^{2}\xi \left(x\right)<E$$


*This corresponds to the energy constraint.*


The above equations are primitive formulae of quantum reliability function theory [46].

#### 6.2.2. Cut-Off Rate

In general, it is difficult to obtain the reliability function. Therefore, it is useful to define and calculate the cut-off rate in the same way as in the discrete system. It is given by the following relation:(98)$$R_{Q}=\max_{0\leq p}\max_{\xi }\mu \left(\xi ,1,p\right)$$
where *s* is fixed to 1.

#### 6.2.3. Example of Cut-Off Rate for Gaussian Channel

In real communication, the only useful quantum states of continuous quantities are coherent states. Let us assume that the coherent-state alphabet is disturbed by Gaussian noise.(99)$$\rho_{0}=\frac{1}{\lambda +1/2}\sum_{n_{p}}{\left(\frac{\lambda -1/2}{\lambda +1/2}\right)}^{n_{p}}\left|n_{p}><n_{p}\right|$$

Even in this case, it is difficult to obtain an exact formula for the reliability function. However, we have the exact solution of the cut-off rate as follows:(100)$$R_{Q}=2\left(\frac{N_{sc}}{2g}+1-D\left(N_{sc}/g\right)\right)+\log D\left(N_{sc}/g\right)$$
where $D\left(N_{sc}\right)=\left(1+\sqrt{N_{sc}^{2}+1}\right)/2$, $N_{sc}$ is the energy of the code word, and $g=\lambda g_{1/2}\left(\lambda \right)$, with(101)$$g_{1/2}\left(t\right)=\frac{1}{2t}\times \frac{{\left(t+1/2\right)}^{1/2}+{\left(t-1/2\right)}^{1/2}}{{\left(t+1/2\right)}^{1/2}-{\left(t-1/2\right)}^{1/2}}$$

In this case, the quantum gain is only the entangled effect of the decision process, i.e., the double quantum gain disappears as in the discrete system.

### 6.3. Importance of Cut-Off Rate and Quantum Advantage

According to reliability function theory, when the rate exceeds the cut-off rate and enters the capacity region, an extremely long code length is required to achieve a sufficiently small error rate. This means increased communication delays and has come to be seen as the biggest drawback of modern communication technology. Therefore, rather than achieving capacity, a coding technique that minimizes code length in the cut-off rate region is essential. Based on the theory in this section, it is preferable to utilize the effect of entangled measurement to shorten code length rather than increase capacity. This is because the amount of information that can be transmitted (bit/s) can be increased by orders of magnitude with optical communication. Therefore, there is no longer any need to forcefully achieve channel capacity, which is efficiency. Rather, delay is the major issue.

In other words, we can point out the following: The primary requirements of modern communication systems are low latency and ultra-high speed. This is because communication systems are fundamentally bidirectional. The challenge for the former lies in shortening the code length, while the latter depends on bandwidth characteristics. Since channel capacity is merely a matter of efficiency, if sufficient bandwidth is available, there is no need to pursue channel capacity. Channel capacity becomes a concern only in inherently unidirectional communication scenarios, such as deep space. Quantum systems, being based on optical communication, have ample bandwidth. However, shortening the code length remains a challenge. Therefore, as pointed out here, it is desirable to develop techniques for shortening code length based on the cut-off rate. Further details will be discussed in the following paper.

## 7. Discovery of a Cipher That Breaks the Shannon Impossibility Theorem

In the previous section, applications of quantum effects to the modern optical communications were considered for achieving high-performance communication based on quantum Shannon theory.

In this section and the sections that follow, we introduce applications of quantum Shannon theory to several technologies, such as cipher and radar. In the case of ciphers, to achieve information-theoretic security for symmetric key ciphers, it is necessary to realize a mechanism that can hide not only data but also secret keys by using quantum noise. This can consist of modulation and demodulation technologies based on the above theories (Figure 8). In this section, we will introduce the basic concept.

### 7.1. New Principle for Ciphers

First of all, we will introduce the application of quantum Shannon theory to the design of ciphers. This is the most exciting application of quantum information science, as it promises to completely outperform conventional functionality in the real world.

There are several no-go theorems in quantum mechanics. Quantum Shannon theory can be considered a theory for designing optimal communication systems that utilize these.

The most important application is to solve the Shannon impossibility theorem, which limits the security of classical cryptography. The Shannon impossibility theorem is given as follows.

**Theorem** **14**({Shannon})**.**
*The information-theoretic security of a symmetric key cipher is limited by the Shannon entropy of the key.*(102)$$H\left(X|Y\right)\leq H\left(K_{S}\right)$$
*where X is the message, $K_{S}$ is the shared secret key, and Y is the ciphertext received by an eavesdropper. The perfect secure cipher is only given by a one-time pad cipher.*

The principle proposed to break the above theorem is the KCQ principle (keyed communication in quantum noise). It was disclosed as a white paper in 2000 and made public in 2003 [47].

**Principle** **1**[{Yuen}]**.**
*The non-orthogonality of a set of signal states can be increased by randomization due to a secret key. One can create a differentiation in reception performance such that the performance of the receiver with a key to randomized quantum states is superior to that of the receiver without the key. That is,*(103)$$\begin{array}{ccc} {\bar{P}}_{e}^{B}\left(with-key\right) & \ll & {\bar{P}}_{e}^{E}\left(without-key\right) \end{array}$$(104)$$\begin{array}{ccc} C^{B}\left(with-key\right) & > & C^{E}\left(without-key\right) \end{array}$$
*where ${\bar{P}}_{e}$ and *C* are the error performance of signal detection and the capacity, respectively. The indexes *B* and *E* mean Bob and Eve. These inequalities are called advantage creation based on KCQ.*

Yuen’s main focus was quantum key distribution (QKD) as an application of his own principle. I introduced Yuen’s concept for lifting the Shannon impossibility theorem of symmetric key ciphers based on the above principle in 2004 (Figure 9) [48]. Since then, our collaborative research has brought various important advances [49,50,51]. In the following, we show a survey of how to apply quantum communication theory to realize a cipher based on KCQ. Its main concept is to use quantum modulation as encryption.

### 7.2. Optical Quantum Modulation as Encryption Based on the New Principle

In this section, we will explain the model of the encryption–decryption scheme by optical quantum modulation–demodulation and clarify the principle behind the appearance of quantum noise effects to hide the data and the shared secret key. We mainly deal with quantum stream ciphers and quantum block cipher formats.

(a) Quantum stream cipher

The first example of ciphers designed according to this principle is the quantum stream cipher. For detailed diagrams of the structure, see references [49,50,51].

Let us denote the protocol, where the data (plaintext) is a sequence of binary signals:(i)The sender prepares a big number (M≫1) of communication bases $\left\{|\alpha_{m}>,|-\alpha_{m}>\right\}$ consisting of two non-orthogonal states (coherent state with high power) such as(105)$$\left\{\right|\alpha_{1}>,|-\alpha_{1}>\},\ldots ,\{|\alpha_{M}>,\left|-\alpha_{M}>\right\}$$
where m={1,2,3,…,M}. One of them is selected by using a pseudorandom number generator (PRNG) with a secret key.(ii)The sender then transmits binary data using the selected binary communication basis.(106)$$0\;or\;1\to Mapper\to |\alpha_{m}>\;or\;|-\alpha_{m}>$$
where Mapper is the random mapping function due to the same PRNG (see [51]).(iii)A receiver who has the same pseudorandom number with the key can identify the communication basis, so they always receive binary signals with small error, because the signal amplitude of binary coherent states is large enough.

A receiver who does not have the key must identify the quantum states of 2M that make up the many communication bases, which increases the error. The reason is that the discrimination error of a multi-quantum state is larger than that of a binary quantum state according to quantum detection theory.

The designer of this cipher could use this error to completely erase the information on the secret key and pseudorandom number structure shared by the sender and receiver, optimizing the scheme based on quantum Shannon theory (see reference [51] for the specific protocol and structure).

As a result, the ciphertext for the eavesdropper is completely masked by quantum error, and it becomes possible to break the Shannon limit (Shannon impossible theorem), which determines the limitation of information-theoretic security of symmetric key ciphers. That is,(107)$$H\left(X|Y^{E_{q}}\right)\gg H\left(K_{S}\right)$$
where $Y^{E_{q}}$ is the ciphertext received by the eavesdropper.

Shannon’s focus was only on ciphertext-only attacks, but in modern times, it is necessary to deal with known-plaintext attacks. For this reason, the following evaluation method has been established [52]. The quantitative evaluation of information-theoretic security is evaluated by the unicity distance for known-plaintext attacks.

**Definition** **11.**
*The unicity distance of known-plaintext attacks for quantum stream ciphers is defined as follows:*

(108)
$$n_{1}^{Q}:H\left(K_{S}|X_{n_{1}^{Q}},Y_{n_{1}^{Q}}^{Eq}\right)=0$$


*It is the minimum value of the ciphertext sequence required for an eavesdropper to be able to guess the key. Then it is given by the following theorem [52].*


**Theorem** **15**({Yuen·Nair})**.**
*The lower bound of the generalized unicity distance for KPA is given as follows:*(109)$$n_{1}^{Q}\geq \frac{|K_{S}|}{C_{1}^{E}},\;C_{1}^{E}=\max_{\left\{\Pi^{E}\right\}}I\left(K^{R};Y^{E_{q}}\right)$$
*where $C_{1}^{E}$ is the maximum amount of mutual information (accessible information) for the eavesdropper’s measurement from a set of quantum states with $K^{R}$ (running key sequence) as a variable. $C_{1}^{E}$ can be replaced with Holevo capacity.*

However, the original idea did not guarantee sufficient security against known-plaintext attacks. In 2007, we solved that problem. That is, one can attain $C_{1}^{E}\to 0$ by generalized randomization [53] (see reference [51] for the detailed concept). In other words, while the one-time pad requires a secret key of the same length as the message, this mechanism can encrypt any message in an information-theoretically secure manner using only a short secret key of a few hundred bits.

As a result, these quantum stream ciphers can provide the encryption system with 100 Gbit/s ∼10 Tbit/s over a communication distance of 1000 km∼10,000 km, guaranteeing information-theoretic security. More details are provided in Section 7.3.

(b) Quantum block cipher

We can consider the block cipher form based on the new principle. Quantum versions of classical block ciphers are called quantum block ciphers. Here we introduce the encryption theory for the quantum block cipher using unitary transformation by Sohma [54,55], which is also a scheme of quantum modulation.

First, let us consider M–ary coherent state composed by *M* blocks of coherent stated as follows:(110)$$\left|\Phi >=\right|\alpha_{1}>|\alpha_{2}>\left|\alpha_{3}>\ldots \right|\alpha_{M}>$$

We consider an operator V that extends the Heisenberg commutation relation for self-adjoint operators on Hilbert spaces to the Weyl–Segal commutation relation.

**Theorem** **16.**
*From the Stone–von Neumann theorem, we can construct the following formula. The quantum characteristic function G(z) for the class of quantum Gaussian state is given as follows:*

(111)
$$\begin{array}{ccc} G(z) & = & TrU|\Phi ><\Phi |U^{†}V\left(z\right)\\ & = & Tr|\Phi ><\Phi |V(L^{T}z) \end{array}$$

*where T means transpose and where*

(112)
$$\begin{array}{ccc} V(z) & = & \exp\{iR^{T}z\} \end{array}$$


(113)
$$\begin{array}{ccc} \;R & = & {\left[\left(q_{1},p_{1}\right),\ldots ,\left(q_{M},p_{M}\right),\right]}^{T} \end{array}$$

*and where $\left(q_{i},p_{i}\right)$ are the canonical conjugate operators. Then L in Equation (Equation 111) is called a symplectic matrix, and it is given by*

(114)
$$\begin{array}{ccc} & & L=\left(\begin{array}{ccc} r_{11}e^{i\theta_{11}} & \ldots & r_{1M}e^{i\theta_{1M}}\\ r_{21}e^{i\theta_{21}} & \ldots & r_{2M}e^{i\theta_{2M}}\\ \vdots & \ldots & \vdots \\ r_{M1}e^{i\theta_{M1}} & \ldots & r_{MM}e^{i\theta_{MM}} \end{array}\right) \end{array}$$


*U in Equation (Equation 111) is called the unitary operator associated with symplectic transformation L, and the following apply:*
*(i)* 
*A set of $L(r_{i,j},\theta_{i,j})$ with different elements is prepared.*
*(ii)* 
*One L is selected from the set using a pseudorandom sequence with the secret key $K_{S}$.*
*(iii)* 
*A quantum ciphertext is generated by a unitary transformation associated with the selected L as follows.*


*Here let us denote a vector of complex amplitudes α of coherent state by*

(115)
$${\vec{\alpha }}_{in}=\left(\alpha_{1},\alpha_{2},\ldots ,\alpha_{M}\right)$$

*then we have the following relation:*

(116)
$${\vec{\alpha }}_{out}=L{\vec{\alpha }}_{in}=\left(\alpha_{1}^{out},\alpha_{2}^{out},\ldots ,\alpha_{M}^{out}\right)$$


*As a result, the unitary transformation for the coherent-state sequence is given as follows:*

(117)
$$U|\Phi >=|\Phi_{out}>=|\alpha_{1}^{out}>\left|\alpha_{2}^{out}>\ldots \right|\alpha_{M}^{out}>$$


*Thus, randomization of complex amplitudes through L converts the basic codes of coherent states into quantum ciphertext with arbitrary complex amplitude by U. When $r_{i,j}=1,\forall i,j$, the above is called phase randomization.*


On the other hand, Bob’s decryption procedure involves applying the inverse $U^{-1}$ of a pseudorandomly selected unitary transformation to the quantum ciphertext by using the same pseudorandom numbers. As a result, a receiver with the key can always receive the basic quantum state code signal before the randomization. For a more detailed explanation, see the study in [56].

### 7.3. Social Implementation

#### 7.3.1. Development of Transceiver for Quantum Stream Cipher

Finally, we will explain the development status of transceivers for practical use of the above encryption. Figure 10 shows the history of the development of the transceiver. In 2002, a handmade prototype was developed, and communication experiments were successfully conducted at 125 Mbps. Subsequently, with the cooperation of Panasonic and Hitachi, a two-way prototype transceiver was developed. Currently, the development of a commercial transceiver for standard quantum stream ciphers has been completed, and we are now in the phase of improving it to generalized quantum stream ciphers with several randomizations.

On the other hand, P. Kumar and his group at Northwestern University independently developed the world’s first high-performance transceiver of quantum stream ciphers in 2002, 2003, and 2009 [57,58,59]. They also demonstrated it for applications in aircraft communication.

#### 7.3.2. Application to Global Optical Network of 100 Gbit/s of Quantum Stream Cipher

When developing new capabilities in cryptography, it is unacceptable to degrade conventional communication performance. To avoid deterioration of communication characteristics, the application of the coherent state, which is the miraculous quantum state, is essential. The cryptographic techniques introduced in this section do not degrade communication characteristics.

This encryption device can be operated without changing the structure of the current global optical network system with optical amplifier repeaters. In other words, encryption of ultra-high-speed optical communications can be completed simply by replacing the current optical transceivers with quantum stream transceivers.

Key distribution for the upper layers of the network utilizes quantum-resistant or PQC, and after key sharing, it can be instantly replaced with a new key using a quantum stream cipher. Thus, we recommend that global network applications in the real world follow the scheme shown in Figure 11. That is, the best solution for defense against today’s urgent cyber attacks is to implement a combined scheme of the quantum-resistant public key cryptography as recommended by the NIST and quantum stream ciphers to protect data [60,61]. Much research and development are being carried out towards these goals [62,63,64,65,66].

On the other hand, the threat of eavesdropping from undersea cables will become extremely important. This is because in undersea communication networks that are approximately 10,000 km long, signals are transmitted from light to light via optical amplifiers. Furthermore, technology has been established to eavesdrop on optical signals from optical amplifier repeaters.

The quantum stream cipher is expected to be a technology that can address this issue, and preliminary experiments have already been carried out [67]. In such systems, keys are pre-installed, and no key distribution is required.

#### 7.3.3. Business for Quantum Stream Cipher Service

Commercialization of quantum stream cipher transceivers and their application systems began in the United States in 2025. Services that meet customer needs will be available within a few years.

## 8. Sensor Applications Beyond the Standard Quantum Limit

### 8.1. Bell State Based on Entangled Coherent State

In general, the Bell states consist of four orthogonal states. However, the four entangled states based on non-orthogonal states are called quasi-Bell states. As a specific example, we present the following states based on coherent states: (118)$$\begin{array}{c} \left\{\begin{array}{ccc} |\Psi_{1}\rangle & = & h_{1}{(|\alpha \rangle }_{A}{|\alpha \rangle }_{B}{+|-\alpha \rangle }_{A}{|-\alpha \rangle }_{B})\\ |\Psi_{2}\rangle & = & h_{2}{(|\alpha \rangle }_{A}{|\alpha \rangle }_{B}{-|-\alpha \rangle }_{A}{|-\alpha \rangle }_{B})\\ |\Psi_{3}\rangle & = & h_{3}{(|\alpha \rangle }_{A}{|-\alpha \rangle }_{B}{+|-\alpha \rangle }_{A}{|-\alpha \rangle }_{B})\\ |\Psi_{4}\rangle & = & h_{4}{(|\alpha \rangle }_{A}{|-\alpha \rangle }_{B}{-|-\alpha \rangle }_{A}{|-\alpha \rangle }_{B}) \end{array}\right. \end{array}$$
where $\left\{h_{i}\right\}$ are constants for normalization, i.e., $h_{1}=h_{3}=1/\sqrt{2(1+\kappa^{2})}$, $h_{2}=h_{4}=1/\sqrt{2(1-\kappa^{2})}$; 〈α|−α〉=κ; and $\langle -\alpha |\alpha \rangle =\kappa^{*}$.

Some of these quasi-Bell states are not orthogonal to each other. Here, if $\kappa =\kappa^{*}$, then their Gram matrix becomes very simple, as follows:(119)$$G=\left(\begin{array}{cccc} 1 & 0 & D & 0\\ 0 & 1 & 0 & 0\\ D & 0 & 1 & 0\\ 0 & 0 & 0 & 1 \end{array}\right)$$
where $D=\frac{2\kappa }{1+\kappa^{2}}$. Let us employ the entanglement of formation as the degree of entanglement. The degrees of entanglement for quasi-Bell states becomes(120)$$\begin{array}{ccc} & & E(|\Psi_{1}\rangle )=E(|\Psi_{3}\rangle )\\ & = & -\frac{1+C_{13}}{2}\log\frac{1+C_{13}}{2}-\frac{1-C_{13}}{2}\log\frac{1-C_{13}}{2} \end{array}$$
where $C_{ij}=\left|\langle \Psi_{i}|\Psi_{j}\rangle \right|$, and we have the special property of $E(|\Psi_{2}\rangle )=E(|\Psi_{4}\rangle )=1$ [68]. Thus $|\Psi_{2}\rangle$ and $|\Psi_{4}\rangle$ have the perfect entanglement.

These physical properties and the mechanisms of teleportation were elucidated by S.J. van Enk [69].

### 8.2. Quantum Reading Scheme

The term of quantum reading as a sensor was pioneered by Pirandola [70] (Appendix B). We will employ the PSK scheme such that the memory on the classical disk consists of the flat and concave areas, which correspond to “0” and “1”, respectively. This type of memory was invented in the beginning of the 1980s, employing a gas laser with high coherence as the light source. Later, its scheme was replaced with a laser diode with high coherence in which the decision scheme consists of the effect of reflection wave injection in laser diodes [71].

For the quantum version of such kind of sensor, one can describe the phase shift by an unitary operator as follows:(121)$$U\left(\theta \right)=exp\left(-\theta a^{†}a\right)$$
where *a* and $a^{†}$ are the annihilation and creation operators for bosonic systems, respectively. Here, the phase factor θ is set to between 0 and 2π.

The reading method in the current PSK scheme is to illuminate the laser light (coherent state) on a disk and to read the phase difference. That is, the signal states of light to discriminate are(122)$$\begin{array}{ccc} |\alpha (0)\rangle & = & I|\alpha \rangle \end{array}$$(123)$$\begin{array}{ccc} \;|\alpha (1)\rangle & = & U(\theta )|\alpha \rangle \end{array}$$
where I is the identity operator.

### 8.3. Error-Free Sensor Applicable to Reaction Control

Let us introduce an application of the quasi-Bell state in reaction control systems [72]. Here we restrict the problem to binary detection. So detection targets are two quantum states. Since the phase difference will be π, the problem is to read the phase shift π from the steady state or input state.

The coherent states are prepared in the current classical memory, and the target signal model becomes(124)$$\begin{array}{ccc} |\alpha (0)\rangle & = & I|\alpha \rangle =|\alpha \rangle \end{array}$$(125)$$\begin{array}{ccc} \;|\alpha (1)\rangle & = & U(\theta =\pi )|\alpha \rangle =|-\alpha \rangle \end{array}$$

If one employs a conventional homodyne receiver to discriminate the above quantum states, the limitation is imposed by the so-called quantum shot noise limit. This is the standard quantum limit in modern information technology.

However, in general, it is reasonable to employ a quantum optimum receiver in order to overcome the standard quantum measurement which is achieved by a homodyne receiver. In this case, from Equation (Equation 14), the limitation depends on the inner product between the two quantum states. For example, the inner product of two coherent states is(126)$$\langle \alpha |-\alpha \rangle =exp\left(-2|\alpha {|}^{2}\right)$$

Let us assume that the $|\Psi_{2}\rangle$ of quasi-Bell states in Equation (Equation 118) is employed as the light source and the *B* mode illuminates a memory disk. The reflection effect $U_{B}\left(\theta \right)$ operates in *B* mode, so the channel model is(127)$$\epsilon_{A⊗B}=I_{A}⊗U_{B}\left(\pi \right)$$

Then, the target states become(128)$$\begin{array}{ccc} |\Psi_{2}\left(0\right)\rangle & = & h_{2}{(|\alpha \rangle }_{A}{|\alpha \rangle }_{B}{-|-\alpha \rangle }_{A}{|-\alpha \rangle }_{B})\\ |\Psi_{2}\left(1\right)\rangle & = & \epsilon_{A⊗B}h_{2}{(|\alpha \rangle }_{A}{|\alpha \rangle }_{B}{-|-\alpha \rangle }_{A}{|-\alpha \rangle }_{B})\\ & = & h_{2}{(|\alpha \rangle }_{A}{|-\alpha \rangle }_{B}{-|-\alpha \rangle }_{A}{|\alpha \rangle }_{B}) \end{array}$$

Thus, the input state $|\Psi_{2}\left(0\right)\rangle =|\Psi_{2}\rangle$ is changed to $|\Psi_{2}\left(1\right)\rangle =|\Psi_{4}\rangle$. The inner product between the above two entangled coherent states in the quasi-Bell state is(129)$$\langle \Psi_{2}\left(0\right)|\Psi_{2}\left(1\right)\rangle =0$$

That is, the inner product becomes zero, and it is independent of the energy of the light source. This is a property of the quasi-Bell state. To check this special property, one can examine the different phase shift as follows:(130)$$|\langle \Psi_{2}\left(0\right)|I_{A}⊗U_{B}\left(\theta \right)|\Psi_{2}\left(0\right)\rangle |>0$$
where θ≠π.

Finally, let us employ the quantum optimum receiver for the binary pure state of two modes. The ultimate detection performance of systems with the coherent state and homodyne, coherent state and quantum receiver, and quasi-Bell state and quantum receiver are given as(131)$$\begin{array}{ccc} {\bar{P}}_{e}\left(C\right) & = & \bar{P}e\left(Homodyne\right) \end{array}$$(132)$$\begin{array}{ccc} \;{\bar{P}}_{e}\left(Q1\right) & = & \frac{1}{2}\left[1-\sqrt{1-4\xi_{0}\xi_{1}e^{\left(-4|\alpha {|}^{2}\right)}}\right] \end{array}$$(133)$$\begin{array}{ccc} {\bar{P}}_{e}\left(Q2\right) & = & 0\; \end{array}$$
respectively. Thus, one can see that the property of the quasi-Bell state provides an attractive improvement, and this property can be obtained only by the combination of the nonclassical state and the quantum optimum receiver. Although there are many nonclassical states, almost all are not changed from the input state to the orthogonal state to the original input state just by reflection. So we interpret that such effectiveness of the quasi-Bell state comes from the special phenomena of entanglement based on non-orthogonal states, which is a feature of quasi-Bell states.

## 9. Conclusions

This paper explains the development of quantum communication theory and its link to Shannon information transmission theory. In particular, it shows how quantum gain appears in the performance of functions such as communications designed using these theories. While quantum applications are currently being actively discussed, new mechanisms that sacrifice some of the conventional performance when applying quantum properties cannot be applied to real-world situations. In other words, quantum communication using qubits, etc., is not suitable for social implementation, because such schemes cannot maintain the current communication capabilities. In addition, we explained that improving delay characteristics (latency) is essential to communication systems in the 21st century. Therefore, it is essential to advantageously apply quantum effects to modern optical communications, which have the highest performance for communication functions. The theory introduced here has been shown to be compatible with such system design. In particular, the quantum stream cipher, as shown in Figure 12, when compared with conventional technology, is expected to provide new possibilities.

Finally, Appendix B, Appendix C and Appendix D discuss paths for future development. In conclusion, readers of this paper will be able to understand the structure of the design principles of ultra-high-speed communication, quantum stream ciphers, and quantum sensors.

## Figures and Tables

**Figure 1 entropy-27-01158-f001:**
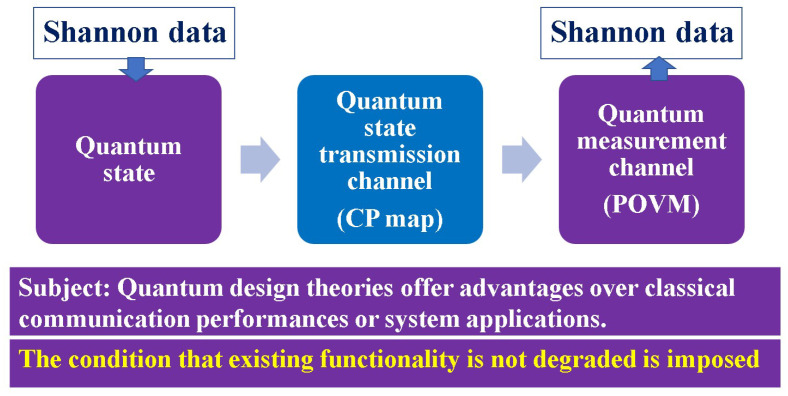
Channel model in quantum Shannon information theory. Shannon information such as digital data or analog signals is transmitted by optical signals with quantum effect governed by a quantum state. The quantum state and quantum measurement determine the communication performance. The existing functionality should not be degraded.

**Figure 2 entropy-27-01158-f002:**
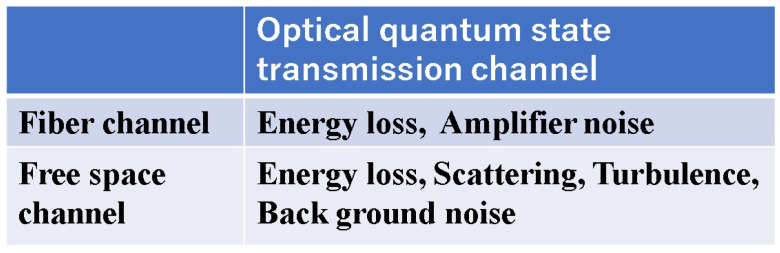
Subject of consideration of quantum state transmission channel. The performance requirements for fiber optic communication are 100 Gbit/s to 100 Tbit/s as the transmission speed and a communication distance of 10,000 km. For terrestrial spatial transmission, the speed is 1 Gbit/s to 10 Gbit/s, and the communication distance is about 10 km. There are no quantum states other than coherent states that satisfy these requirements. This is a consequence of Theorem 1.

**Figure 3 entropy-27-01158-f003:**
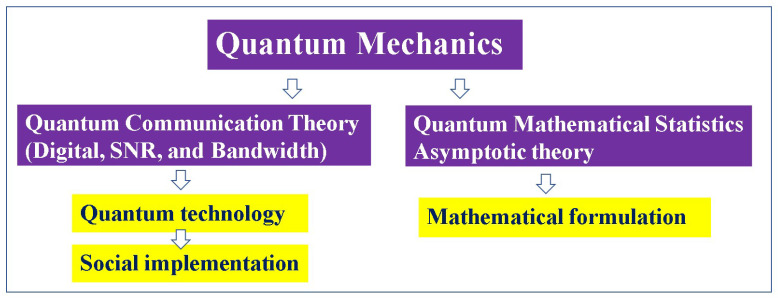
The fundamental concepts of quantum communication theory and quantum mathematical statistics are different, though they have certain similarities. The former has to guarantee operational meaning for quantum communication technology. The latter provides a formulation for statistics, with asymptotic properties, etc., as its main objective. Thus, it does not provide operational meaning. The relationship between the two is the same in classical theory.

**Figure 4 entropy-27-01158-f004:**
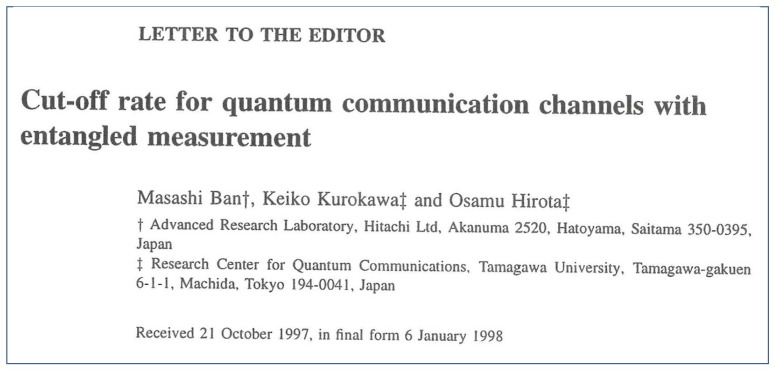
This paper clearly showed the specific structure of definition and effect of cut-off rate of the decision operator based on entangled measurement [13].

**Figure 5 entropy-27-01158-f005:**
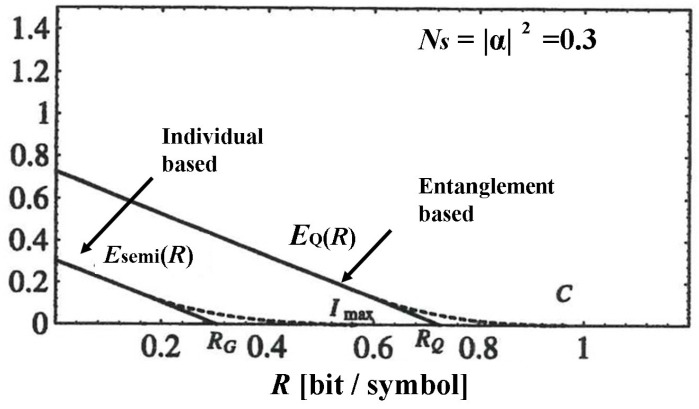
Numerical example of reliability function and cut-off rate for the decision operators based on entangled measurement and individual measurement [45]. This shows the advantage of the reliability functions and cut-off rate based on entangled measurement.

**Figure 6 entropy-27-01158-f006:**
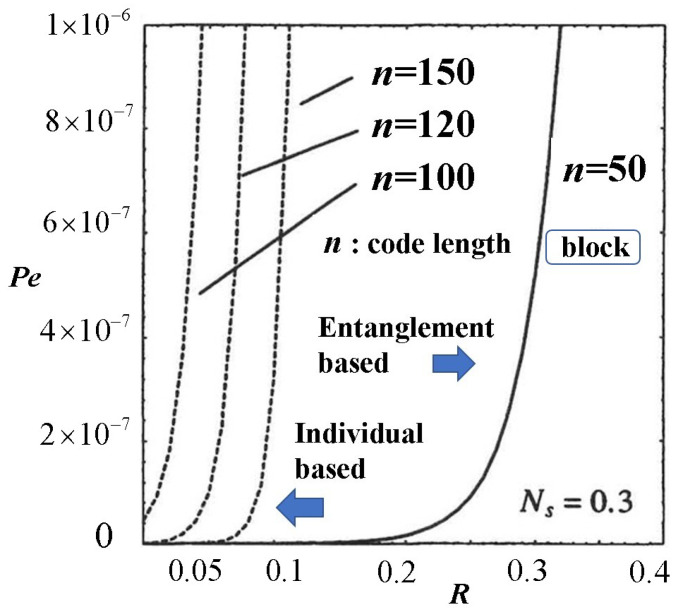
Numerical examples of quantum advantage according to block code length by decision operators based on entangled measurement and individual measurement [45].

**Figure 7 entropy-27-01158-f007:**
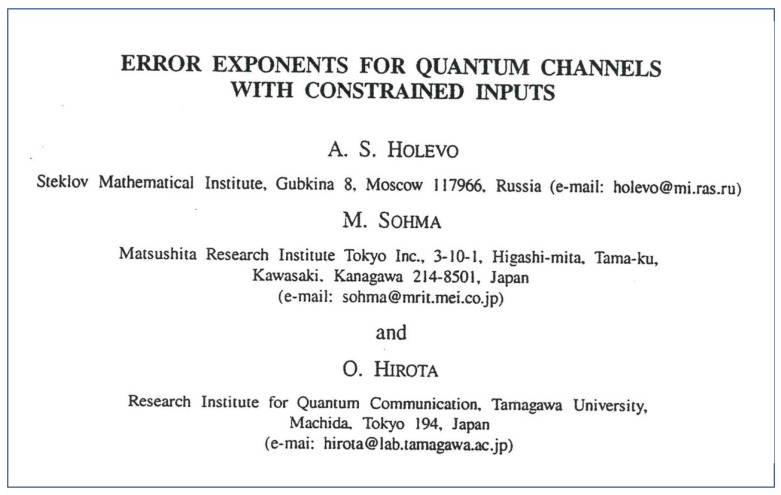
This paper provides the first rigorous definition of reliability function and cut-off rate for infinite alphabet systems under the energy constraint [46]. Especially, the concrete form of quantum cut-off rate is given.

**Figure 8 entropy-27-01158-f008:**
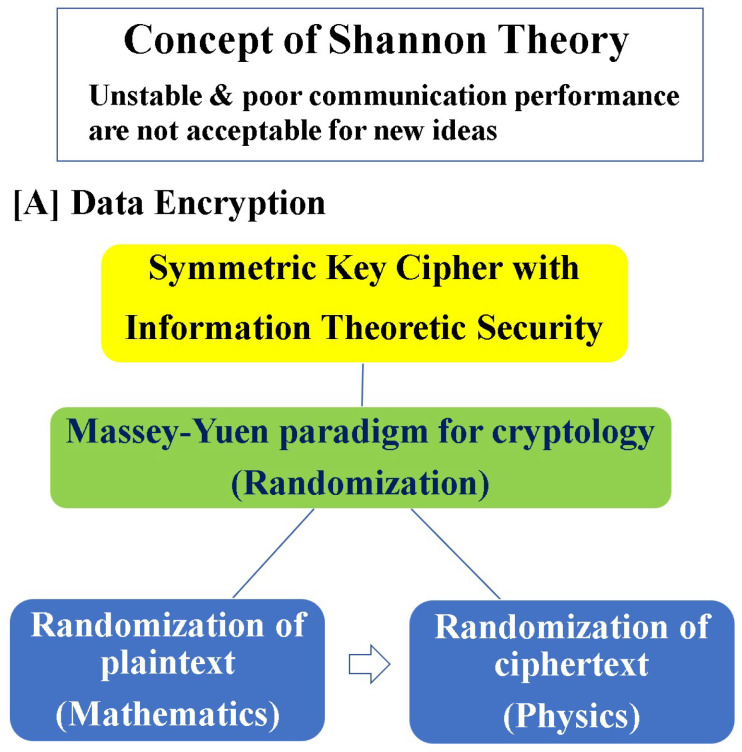
Basic scenario in the Shannon concept for cryptology. One can obtain the drastic quantum advantage in physical encryption based on the quantum modulation–demodulation protocol.

**Figure 9 entropy-27-01158-f009:**
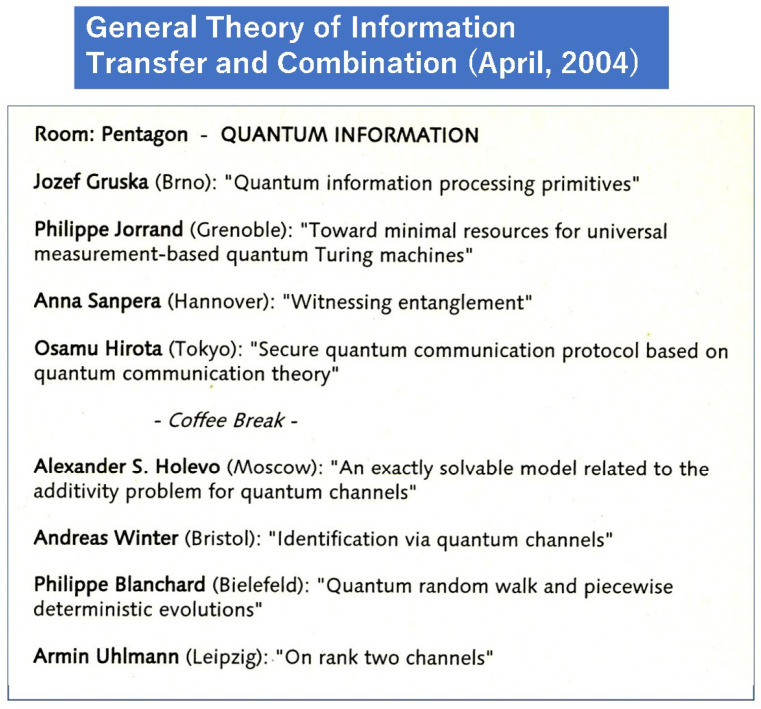
Program of the Conference on General theory of Information Transfer and Combination held at the Center for Interdisciplinary Research, Bielefeld University, organized by Prof. Rudolf Ahlswede.

**Figure 10 entropy-27-01158-f010:**
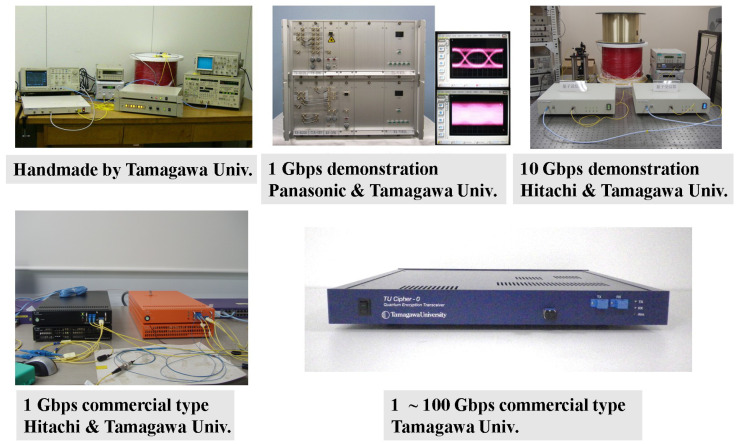
History of the development of quantum stream cipher communication devices, which enable encrypted communication of arbitrary ultra-high-speed binary data with randomized coherent-state signals without delay. Setup is complete by simply replacing the optical transceiver currently in use with the above.

**Figure 11 entropy-27-01158-f011:**
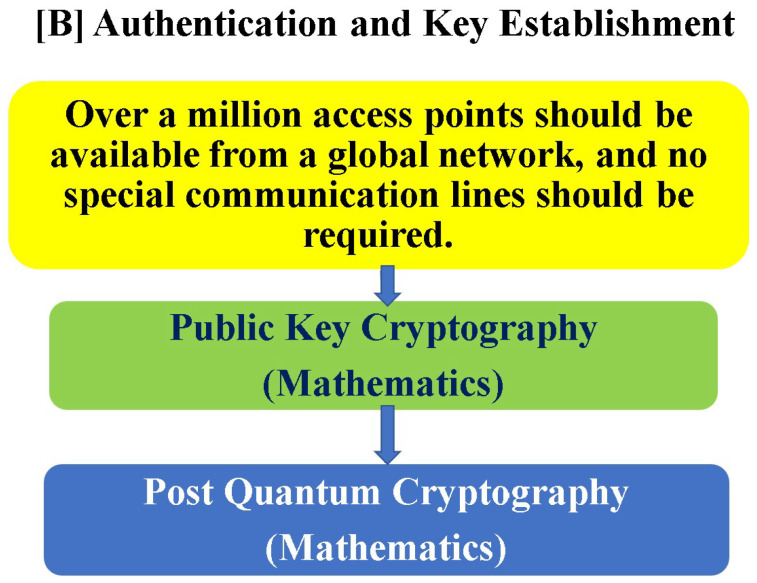
Key distribution and authentication require strong conditions on utility for global networks. After key establishment, the key is exchanged for a new one by using the information-theoretically secure symmetric key cipher described above. It is also safe even if the correct key is leaked after the encrypted communication.

**Figure 12 entropy-27-01158-f012:**
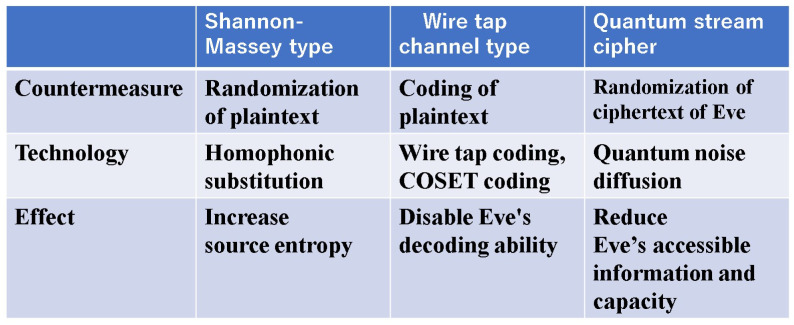
Comparison of the characteristics of the conventional information-theoretic cipher and the quantum stream cipher. Quantum stream ciphers evolved from forms of Shannon-Massey ciphers.

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
