# Peer review of "Quantum Shannon Information Theory—Design of Communication, Ciphers, and Sensors"

_entropy, 2025, doi:10.3390/e27111158_

Round 1

Reviewer 1 Report

Comments and Suggestions for Authors
  1. In this article the author described the methodology, techniques, and tools in a very detailed fashion, which makes the subject material presented in this paper more comprehensive.
  2. The computations presented in this paper are correct and a new addition to the corresponding literature. 
  3. The obtained results are correct, and the presentation of the paper is really good.
  4. The reference list is complete.

Based on the above-mentioned comments, I strongly recommend publication of this paper in a present format.

Reviewer 2 Report

Comments and Suggestions for Authors

This is a comprehensive review paper that bridges foundational quantum information theory with practical, real-world communication systems. It has a strong perspective, arguing for the importance of operational meaning and compatibility with existing high-performance optical infrastructure.

Based on a detailed reading, here is a list of suggestions for improvement:

  1. Standardize the reference list according to the journal's specific guidelines. Verify all arXiv links and dates.
  2. The paper repeatedly claims to present a "perfectly secure cipher." This is a very strong claim that requires careful qualification. The security proofs for these KCQ (Y-00) type ciphers are based on specific noise models and assumptions about the eavesdropper's technological capabilities (e.g., they cannot perform joint measurements over long code-words, a standard assumption in quantum cryptography). It is not "perfect" in the information-theoretic sense of a one-time pad, which is secure against any attacker. Consider rephrasing to be more precise. For example: "…realizes ciphers with provable security based on quantum noise," or "…which can achieve information-theoretic security under realistic physical-layer assumptions." A footnote or clarification in the text explaining the security model would be beneficial.
  3. The paper positions the quantum stream cipher as "overcoming" or "lifting" Shannon's impossibility theorem. This is rhetorically powerful but can be misleading. A more accurate interpretation is that they are operating in a different physical framework. Shannon's theorem assumes a classical channel; by using a quantum mechanical channel, the rules change, and new possibilities (like hiding the key in quantum noise) emerge. They are not violating a mathematical theorem but rather changing its underlying assumptions. I warmly advise to clarify this point. A phrase like "…by exploiting quantum mechanical properties not considered in the classical Shannon framework, we can circumvent the limitations imposed by the classical Shannon impossibility theorem" would be more accurate.
  4. Page 5, Definition 3: "In some cases, X(r) is called the positive operator valued measure (POVM)." is somehow unclear. Which cases?

Misprints and Typos

Page 2: "Shumacher" is a frequent typo for "Schumacher" (e.g., in references 4, 6).

Page 5, Remark 2: "by observer" should be "by the observer".

Page 6, Section 3.3: "degree" is perhaps"dimension" or "extension" ?

Page 9, Section 4.1.4: "measurment" -> "measurement".

Page 10, Section 4.2.1: "it has still certain difficulty" -> "it still has certain difficulties".

Page 10, Section 4.2.1:  as denoted in the following. --> as discussed  in the following.

Page 11, just above Eq. (35) "ignor" -> "ignore".

Page 15, Figure 3: "old style" is informal. Consider making it more clear.

Page 21, Eq. (83) The state vectors description is obscure are typeset incorrectly

Page 28, Eq(107): The notation \(Y^{E_q}\) is introduced here but not used consistently elsewhere (often just Y).

Page 29, Eq(111): \(U^TV(z)\) -> The T superscript is unclear. Should this be a dagger or is really transpose?

Page 30, Social Implementation: "Panasonic" is mentioned. It's worth checking if this is the correct corporate name (formerly Matsushita).

Page 33, Eq(127): The symbol \(\bullet_{A\otimes B}\) is used instead of the defined channel \(\Phi_{A\otimes B}\) or similar. Is this the exact meaning?

Page 38, Appendix D: "a real systems" -> "real-world systems".

The transition from the theoretical foundations (Sections 2-5) to the applications (Sections 6-8) is good. However, Section 6 (Examples) feels a bit like a collection of results. A stronger concluding sentence for Section 6.3, summarizing the key takeaway about delay vs. capacity, would improve the flow.

The text frequently says "See Fig.X" but does not always describe what the figure shows. For example, before Figure 5 and 6, a brief description like "Figure 5 shows the numerical comparison of the reliability functions for individual and entangled measurements, demonstrating a clear quantum advantage" would be helpful.

Round 2

Reviewer 2 Report

Comments and Suggestions for Authors

Properly revised and proper rebuttal to point 2/3.